# Covariant Compositional Networks For Learning Graphs

**Risi Kondor, Truong Son Hy, Horace Pan & Brandon M. Anderson**
Department of Computer Science
The University of Chicago
Chicago, IL - 60637
`{risi,hytruongson,hopan,brandona}@cs.uchicago.edu`

**Shubhendu Trivedi**
Toyota Technological Institute
Chicago, IL - 60637
`shubhendu@ttic.edu`

## Abstract

Most existing neural networks for learning graphs address permutation invariance by conceiving of the network as a message passing scheme, where each node sums the feature vectors coming from its neighbors. We argue that this imposes a limitation on their representation power, and instead propose a new general architecture for representing objects consisting of a hierarchy of parts, which we call *covariant compositional networks* (CCNs). Here, covariance means that the activation of each neuron must transform in a specific way under permutations, similarly to steerability in CNNs. We achieve covariance by making each activation transform according to a tensor representation of the permutation group, and derive the corresponding tensor aggregation rules that each neuron must implement. Experiments show that CCNs can outperform competing methods on standard graph learning benchmarks.

## 1 Introduction

Learning on graphs has a long history in the kernels literature, including approaches based on random walks (Gärtner, 2002; Borgwardt & Kriegel, 2005; Feragen et al., 2013), counting subgraphs (Shervashidze et al., 2009), spectral ideas (Vishwanathan et al., 2010), label propagation schemes with hashing (Shervashidze et al., 2011; Neumann et al., 2016), and even algebraic ideas (Kondor & Borgwardt, 2008). Many of these papers address moderate size problems in chemo- and bioinformatics, and the way they represent graphs is essentially fixed.

Recently, with the advent of deep learning and much larger datasets, a sequence of neural network based approaches have appeared to address the same problem, starting with (Scarselli et al., 2009). In contrast to the kernels framework, neural networks effectively integrate the classification or regression problem at hand with learning the graph representation itself, in a single, end-to-end system. In the last few years, there has been a veritable explosion in research activity in this area. Some of the proposed graph learning architectures (Duvenaud et al., 2015; Kearnes et al., 2016; Niepert et al., 2016) directly seek inspiration from the type of classical CNNs that are used for image recognition (LeCun et al., 1998; Krizhevsky et al., 2012). These methods involve first fixing a vertex ordering, then moving a filter across vertices while doing some computation as a function of the local neighborhood to generate a representation. This process is then repeated multiple times like in classical CNNs to build a deep graph representation. Other notable works on graph neural networks include (Li et al., 2016; Schütt et al., 2017; Battaglia et al., 2016; Kipf & Welling, 2017). Very recently, (Gilmer et al., 2017) showed that many of these approaches can be seen to be specific instances of a general message passing formalism, and coined the term *message passing neural networks* (MPNNs) to refer to them collectively.

While MPNNs have been very successful in applications and are an active field of research, they differ from classical CNNs in a fundamental way: the internal feature representations in CNNs are *equivariant* to such transformations of the inputs as translation and rotations (Cohen & Welling, 2016; 2017), the internal representations in MPNNs are fully invariant. This is a direct result of the fact that MPNNs deal with the permutation invariance issue in graphs simply by summing the messages coming from each neighbor. In this paper we argue that this is a serious limitation that restricts the representation power of MPNNs.

MPNNs are ultimately compositional (part-based) models, that build up the representation of the graph from the representations of a hierarchy of subgraphs. To address the covariance issue, we study the covariance behavior of such networks in general, introducing a new general class of neural network architectures, which we call *compositional networks* (comp-nets). One advantage of this generalization is that instead of focusing attention on the mechanics of how information propagates from node to node, it emphasizes the connection to convolutional networks, in particular, it shows that what is missing from MPNNs is essentially the analog of *steerability*.

Steerability implies that the activations (feature vectors) at a given neuron must transform according to a specific representation (in the algebraic sense) of the symmetry group of its receptive field, in our case, the group of permutations, $\mathbb{S}_m$. In this paper we only consider the defining representation and its tensor products, leading to first, second, third etc. order *tensor activations*. We derive the general form of covariant tensor propagation in comp-nets, and find that each "channel" in the network corresponds to a specific way of contracting a higher order tensor to a lower order one. Note that here by *tensor activations* we mean not just that each activation is expressed as a multidimensional array of numbers (as the word is usually used in the neural networks literature), but also that it transforms in a specific way under permutations, which is a more stringent criterion. The parameters of our covariant comp-nets are the entries of the mixing matrix that prescribe how these channels communicate with each other at each node. Our experiments show that this new architecture can beat scalar message passing neural networks on several standard datasets.

## 2 LEARNING GRAPHS

Graph learning encompasses a broad range of problems where the inputs are graphs and the outputs are class labels (classification), real valued quantities (regression) or more general, possibly combinatorial, objects. In the standard supervised learning setting this means that the training set consists of $m$ input/output pairs $\{(G_1, y_1), (G_2, y_2), \ldots, (G_m, y_m)\}$, where each $G_i$ is a graph and $y_i$ is the corresponding label, and the goal is to learn a function $h: G \to y$ that will successfully predict the labels of further graphs that were not in the training set.

By way of fixing our notation, in the following we assume the each graph $G$ is a pair $(V, E)$, where $V$ is the vertex set of $G$ and $E \subseteq V \times V$ is its edge set. For simplicity, we assume that $V = \{1, 2, \ldots, n\}$. We also assume that $G$ has no self-loops ($(i, i) \notin E$ for any $i \in V$) and that $G$ is symmetric, i.e., $(i, j) \in E \Rightarrow (j, i) \in E$[1]. We will, however, allow each edge $(i, j)$ to have a corresponding weight $w_{i,j}$, and each vertex $i$ to have a corresponding feature vector (vertex label) $l_i \in \mathbb{R}^d$. The latter, in particular, is important in many scientific applications, where $l_i$ might encode, for example, what type of atom occupies a particular site in a molecule, or the identity of a protein in a biochemical interaction network. All the topological information about $G$ can be summarized in an adjacency matrix $A \in \mathbb{R}^{n \times n}$, where $A_{i,j} = w_{i,j}$ if $i$ and $j$ are connected by an edge, and otherwise $A_{i,j} = 0$. When dealing with labeled graphs, we also have to provide $(l_1, \ldots, l_n)$ to fully specify $G$.

One of the most fascinating aspects of graphs, but also what makes graph learning challenging, is that they involve structure at multiple different scales. In the case when $G$ is the graph of a protein, for example, an ideal graph learning algorithm would represent $G$ in a manner that simultaneously captures structure at the level of individual atoms, functional groups, interactions between functional groups, subunits of the protein, and the protein's overall shape.

The other major requirement for graph learning algorithms relates to the fact that the usual ways to store and present graphs to learning algorithms have a critical spurious symmetry: If we were to

---

[1]Our framework has natural generalizations to non-symmetric graphs and graphs with self-loops, but in the interest of keeping our discussion as simple as possible, we will not discuss these cases in the present paper.

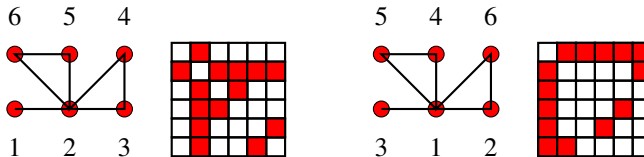

Figure 1: (a) A small graph $G$ with 6 vertices and its adjacency matrix. (b) An alternative form $G'$ of the same graph, derived from $G$ by renumbering the vertices by a permutation $\sigma\colon \{1, 2, \ldots, 6\} \mapsto \{1, 2, \ldots, 6\}$. The adjacency matrices of $G$ and $G'$ are different, but topologically they represent the same graph. Therefore, we expect the feature map $\phi$ to satisfy $\phi(G) = \phi(G')$.

permute the vertices of $G$ by any permutation $\sigma\colon \{1, 2, \ldots, n\} \to \{1, 2, \ldots, n\}$ (in other words, rename vertex 1 as $\sigma(1)$, vertex 2 as $\sigma(2)$, etc.), then the adjacency matrix would change to

$$A'_{i,j} = A_{\sigma^{-1}(i), \sigma^{-1}(j)},$$

and simultaneously the vertex labels would change to $(l'_1, \ldots, l'_n)$, where $l'_i = l_{\sigma^{-1}(i)}$. However, $G' = (A', l'_1, \ldots, l'_n)$ would still represent exactly the same graph as $G = (A, l_1, \ldots, l_n)$. In particular, (a) in training, whether $G$ or $G'$ is presented to the algorithm must not make a difference to the final hypothesis $h$ that it returns, (b) $h$ itself must satisfy $h(G) = h(G')$ for any labeled graph and its permuted variant.

Most learning algorithms for combinatorial objects hinge on some sort of fixed or learned internal representation of data, called the *feature map*, which, in our case we denote $\phi(G)$. The set of all $n!$ possible permutations of $\{1, 2, \ldots, n\}$ forms a group called the symmetric group of order $n$, denoted $\mathbb{S}_n$. The permutation invariance criterion can then be formulated as follows (Figure 1).

**Definition 1.** *Let $\mathcal{A}$ be a graph learning algorithm that uses a feature map $G \mapsto \phi(G)$. We say that the feature map $\phi$ (and consequently the algorithm $\mathcal{A}$) is **permutation invariant** if, given any $n \in \mathbb{N}$, any $n$ vertex labeled graph $G = (A, l_1, \ldots, l_n)$, and any permutation $\sigma \in \mathbb{S}_n$, letting $G' = (A', l'_1, \ldots, l'_n)$, where $A'_{i,j} = A_{\sigma^{-1}(i), \sigma^{-1}(j)}$ and $l'_i = l_{\sigma^{-1}(i)}$, we have that $\phi(G) = \phi(G')$.*

Capturing multiscale structure and respecting permutation invariance are the two the key constraints around which most of the graph learning literature revolves. In kernel based learning, for example, invariant kernels have been constructed by counting random walks (Gärtner, 2002), matching eigenvalues of the graph Laplacian (Vishwanathan et al., 2010) and using algebraic ideas (Kondor & Borgwardt, 2008).

## 3 Compositional networks

Many recent graph learning papers, whether or not they make this explicit, employ a *compositional* approach to modeling graphs, building up the representation of $G$ from representations of subgraphs. At a conceptual level, this is similar to part-based modeling, which has a long history in machine learning (Fischler & Elschlager, 1973; Ohta et al., 1978; Tu et al., 2005; Felzenszwalb & Huttenlocher, 2005; Zhu & Mumford, 2006; Felzenszwalb et al., 2010). In this section we introduce a general, abstract architecture called **compositional networks (comp-nets)** for representing complex objects as a combination of their parts, and show that several exisiting graph neural networks can be seen as special cases of this framework.

**Definition 2.** *Let $\mathcal{G}$ be a compound object with $n$ elementary parts (atoms) $\mathcal{E} = \{e_1, \ldots, e_n\}$. A **composition scheme** for $\mathcal{G}$ is a directed acyclic graph (DAG) $\mathcal{M}$ in which each node $\mathfrak{n}_i$ is associated with some subset $\mathcal{P}_i$ of $\mathcal{E}$ (these subsets are called the **parts** of $\mathcal{G}$) in such a way that*

1. *If $\mathfrak{n}_i$ is a leaf node, then $\mathcal{P}_i$ contains a single atom $e_{\xi(i)}$[2].*
2. *$\mathcal{M}$ has a unique root node $\mathfrak{n}_r$, which corresponds to the entire set $\{e_1, \ldots, e_n\}$.*
3. *For any two nodes $\mathfrak{n}_i$ and $\mathfrak{n}_j$, if $\mathfrak{n}_i$ is a descendant of $\mathfrak{n}_j$, then $\mathcal{P}_i \subset \mathcal{P}_j$.*

We define a compositional network as a composition scheme in which each node $\mathfrak{n}_i$ also carries a *feature vector $f_i$* that provides a representation of the corresponding part (Figure 2). When we want

---

[2]Here $\xi$ is just a function that establishes the mapping between each leaf node and the corresponding atom.

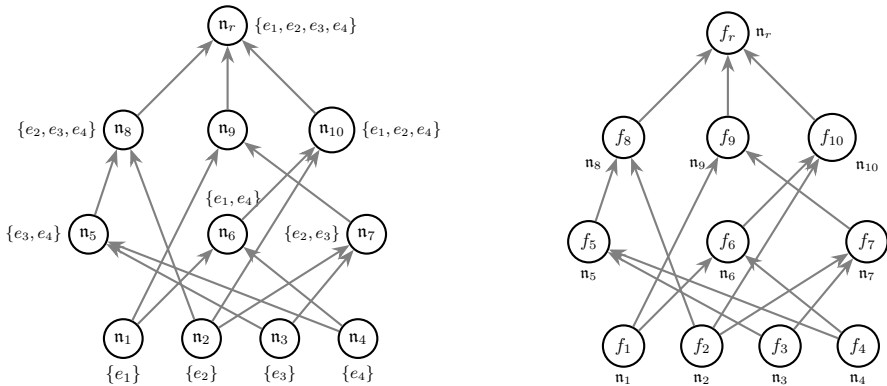

Figure 2: (a) A *composition scheme* for an object $\mathcal{G}$ is a DAG in which the leaves correspond to atoms, the internal nodes correspond to sets of atoms, and the root corresponds to the entire object. (b) A *compositional network* is a composition scheme in which each node $\mathfrak{n}_i$ also carries a feature vector $f_i$. The feature vector at $\mathfrak{n}_i$ is computed from the feature vectors of the children of $\mathfrak{n}_i$.

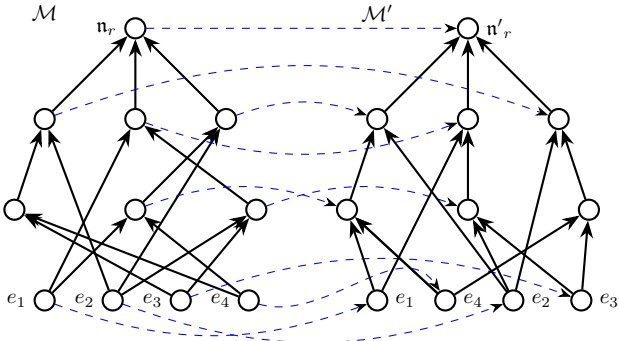

Figure 3: A minimal requirement for composition schemes is that they be invariant to permutation, i.e. that if the numbering of the atoms is changed by a permutation $\sigma$, then we must get an isomorphic DAG. Any node in the new DAG that corresponds to $\{e'_{i_1}, \ldots, e'_{i_k}\}$ must have a corrresponding node in the old DAG corresponding to $\{e_{\sigma^{-1}(i_1)}, \ldots, e_{\sigma^{-1}(i_k)}\}$.

to emphasize the connection to more classical neural architectures, we will refer to $\mathfrak{n}_i$ as the $i$'th **neuron**, $\mathcal{P}_i$ as its **receptive field**[3], and $f_i$ as its **activation**.

**Definition 3.** *Let $\mathcal{G}$ be a compound object in which each atom $e_i$ carries a label $l_i$, and $\mathcal{M}$ a composition scheme for $\mathcal{G}$. The corresponding **compositional network** $\mathcal{N}$ is a DAG with the same structure as $\mathcal{M}$ in which each node $\mathfrak{n}_i$ also has an associated feature vector $f_i$ such that*

1. *If $\mathfrak{n}_i$ is a leaf node, then $f_i = l_{\xi(i)}$.*
2. *If $\mathfrak{n}_i$ is a non-leaf node, and its children are $\mathfrak{n}_{c_1}, \ldots, \mathfrak{n}_{c_k}$, then $f_i = \Phi(f_{c_1}, f_{c_2}, \ldots, f_{c_k})$ for some* ***aggregation function*** *$\Phi$. (Note: in general, $\Phi$ can also depend on the relationships between the subparts, but for now, to keep the discussion as simple as possible, we ignore this possibility.)*

*The representation $\phi(\mathcal{G})$ afforded by the comp-net is given by the feature vector $f_r$ of the root.*

Note that while, for the sake of concreteness, we call the $f_i$'s "feature vectors", there is no reason a priori why they need to be vectors rather than some other type of mathematical object. In fact, in the second half of the paper we make a point of treating the $f_i$'s as tensors, because that is what will make it the easiest to describe the specific way that they transform with respect to permutations.

---

[3] Here and in the following by the "receptive field" of a neuron $\mathfrak{n}_i$ in a feed-forward network we mean the set of all input neurons from which information can propagate to $\mathfrak{n}_i$.

In compositional networks for graphs, the atoms will usually be the vertices, and the $\mathcal{P}_i$ parts will correspond to clusters of nodes or neighborhoods of given radii. Comp-nets are particularly attractive in this domain because they can combine information from the graph at different scales. The comp-net formalism also suggests a natural way to satisfy the permutation invariance criterion of Definition 1.

**Definition 4.** *Let $\mathcal{M}$ be the composition scheme of an object $\mathcal{G}$ with $n$ atoms and $\mathcal{M}'$ the composition scheme of another object that is equivalent in structure to $\mathcal{G}$, except that its atoms have been permuted by some permutation $\sigma \in \mathbb{S}_n$ ($e_i' = e_{\sigma^{-1}(i)}$ and $\ell_i' = \ell_{\sigma^{-1}(i)}$). We say that $\mathcal{M}$ (more precisely, the algorithm generating $\mathcal{M}$) is **permutation invariant** if there is a bijection $\psi\colon \mathcal{M} \to \mathcal{M}'$ taking each $\mathfrak{n}_a \in \mathcal{M}$ to some $\mathfrak{n}_b' \in \mathcal{M}'$ such that if $\mathcal{P}_a = \{e_{i_1}, \ldots, e_{i_k}\}$, then $\mathcal{P}_b' = \{e_{\sigma(i_1)}', \ldots, e_{\sigma(i_k)}'\}$.*

**Proposition 1.** *Let $\phi(\mathcal{G})$ be the output of a comp-net based on a composition scheme $\mathcal{M}$. Assume*
1. *$\mathcal{M}$ is permutation invariant in the sense of Definition 4.*
2. *The aggregation function $\Phi(f_{c_1}, f_{c_2}, \ldots, f_{c_k})$ used to compute the feature vector of each node from the feature vectors of its children is invariant to the permutations of its arguments.*

*Then the overall representation $\phi(\mathcal{G})$ is invariant to permutations of the atoms. In particular, if $\mathcal{G}$ is a graph and the atoms are its vertices, then $\phi$ is a permutation invariant graph representation.*

### 3.1 MESSAGE PASSING NEURAL NETWORKS AS A SPECIAL CASE OF COMP-NETS

Graph learning is not the only domain where invariance and multiscale structure are important: the most commonly cited reasons for the success of convolutional neural networks (CNNs) in image tasks is their ability to address exactly these two criteria in the vision context. Furthermore, each neuron $\mathfrak{n}_i$ in a CNN aggregates information from a small set of neurons from the previous layer, therefore its receptive field, corresponding to $\mathcal{P}_i$, is the union of the receptive fields of its "children", so we have a hierarchical structure very similar to that described in the previous section. In this sense, CNNs are a specific kind of compositional network, where the atoms are pixels. This connection has inspired several authors to frame graph learning as a generalization of convolutional nets to the graph domain (Bruna et al., 2014; Henaff et al., 2015; Duvenaud et al., 2015; Defferrard et al., 2016; Kipf & Welling, 2017). While in mathematics convolution has a fairly specific meaning that is side-stepped by this analogy, the CNN analogy does suggest that a natural way to define the $\Phi$ aggregation functions is to let $\Phi(f_{c_1}, f_{c_2}, \ldots, f_{c_k})$ be a linear function of $f_{c_1}, f_{c_2}, \ldots, f_{c_k}$ followed by a pointwise nonlinearity, such as a ReLU operation.

To define a comp-net for graphs we also need to specify the composition scheme $\mathcal{M}$. Many algorithms define $\mathcal{M}$ in layers, where each layer (except the last) has one node for each vertex of $G$:

$\mathcal{M}$1. In layer $\ell = 0$ each node $\mathfrak{n}_i^0$ represents the single vertex $\mathcal{P}_i^0 = \{i\}$.

$\mathcal{M}$2. In layers $\ell = 1, 2, \ldots, L$, node $\mathfrak{n}_i^\ell$ is connected to all nodes from the previous level that are neighbors of $i$ in $G$, i.e., the children of $\mathfrak{n}_i^\ell$ are

$$\text{ch}(\mathfrak{n}_i^\ell) = \{\, \mathfrak{n}_j^{\ell-1} \mid j \in \mathcal{N}(i) \,\},$$

where $\mathcal{N}(i)$ denotes the set of neighbors of $i$ in $G$. Therefore, $\mathcal{P}_i^\ell = \bigcup_{j \in \mathcal{N}(i)} \mathcal{P}_j^{\ell-1}$.

$\mathcal{M}$3. In layer $L{+}1$ we have a single node $\mathfrak{n}_r$ that represents the entire graph and collects information from all nodes at level $L$.

Since this construction only depends on topological information about $G$, the resulting composition scheme is guaranteed to be permutation invariant in the sense of Definition 4.

A further important consequence of this way of defining $\mathcal{M}$ is that the resulting comp-net can be equivalently interpreted as *label propagation algorithm*, where in each round $\ell = 1, 2, \ldots, L$, each vertex aggregates information from its neighbors and then updates its own label.

---

**Algorithm 1** The label propagation algorithm corresponding to $\mathcal{M}1$–$\mathcal{M}3$

```
for each vertex i
    f_i^0 ← l_i
for ℓ = 1 to L
    for each vertex i
        f_i^ℓ ← Φ(f_{i_1}^{ℓ-1}, ..., f_{i_k}^{ℓ-1})  where  N(i) = {i_1, ..., i_k}
φ(G) ≡ f_r ← Φ(f_1^L, ..., f_n^L)
```

---

Many authors choose to describe graph neural networks exclusively in terms of label propagation, without mentioning the compositional aspect of the model. Gilmer et al. (2017) call this general approach *message passing neural networks*, and point out that a range of different graph learning architectures are special cases of it. More broadly, the classic Weisfeiler–Lehman test of isomorphism also follows the same logic (Weisfeiler & Lehman, 1968; Read & Corneil, 1977; Cai et al., 1992), and so does the related Weisfeiler–Lehman kernel, arguably the most successful kernel-based approach to graph learning (Shervashidze et al., 2011). Note also that in label propagation or message passing algorithms there is a clear notion of the *source domain* of vertex $i$ at round $\ell$, as the set of vertices that can influence $f_i^\ell$, and this corresponds exactly to the receptive field $\mathcal{P}_i^\ell$ of "neuron" $\mathfrak{n}_i^\ell$ in the comp-net picture.

The following proposition is immediate from the form of Algorithm 1 and reassures us that message passing neural networks, as special cases of comp-nets, do indeed produce permutation invariant representations of graphs.

**Proposition 2.** *Any label propagation scheme in which the aggregation function $\Phi$ is invariant to the permutations of its arguments is invariant to permutations in the sense of Definition 1.*

In the next section we argue that invariant message passing networks are limited in their representation power, however, and describe a generalization via comp-nets that overcomes some of these limitations.

## 4 COVARIANT COMPOSITIONAL NETWORKS

One of the messages of the present paper is that invariant message passing algorithms, of the form described in the previous section, are *not* the most general possible compositional models for producing permutation invariant representations of graphs (or of compound objects, in general).

Once again, an analogy with image recognition is helpful. Classical CNNs face two types of basic image transformations: translations and rotations. With respect to translations (barring pooling, edge effects and other complications), CNNs behave in a quasi-invariant way, in the sense that if the input image is translated by any integer amount $(t_x, t_y)$, the activations in each layer $\ell = 1, 2, \ldots L$ translate the same way: the activation of any neuron $\mathfrak{n}_{i,j}^\ell$ is simply transferred to neuron $\mathfrak{n}_{i+t_1,j+t_2}^\ell$, i.e., $f'^\ell_{i+t_1,j+t_2} = f^\ell_{i,j}$. This is the simplest manifestation of a well studied property of CNNs called *equivariance* (Cohen & Welling, 2016; Worrall et al., 2017).

With respect to rotations, however, the situation is more complicated: if we rotate the input image by, e.g., 90 degrees, not only will the part of the image that fell in the receptive field of a particular neuron $\mathfrak{n}_{i,j}^\ell$ move to the receptive field of a different neuron $\mathfrak{n}_{j,-i}^\ell$, but the orientation of the receptive field will also change (Figure 4). Consequently, features which were, for example, previously picked up by horizontal filters will now be picked up by vertical filters. Therefore, in general, $f'^\ell_{j,-i} \neq f^\ell_{i,j}$. It can be shown that one cannot construct a CNN for images that behaves in a quasi-invariant way with respect to both translations and rotations unless every filter is directionless.

It is, however, possible to construct a CNN in which the activations transform in a predictable and reversible way, in particular, $f'^\ell_{j,-i} = R(f^\ell_{i,j})$ for some fixed invertible function $R$. This phenomenon is called *steerability*, and has a significant literature in both classical signal processing (Freeman & Adelson, 1991; Simoncelli et al., 1992; Perona, 1995; Teo & Hel-Or, 1998; Manduchi et al., 1998) and the neural networks field (Cohen & Welling, 2017).

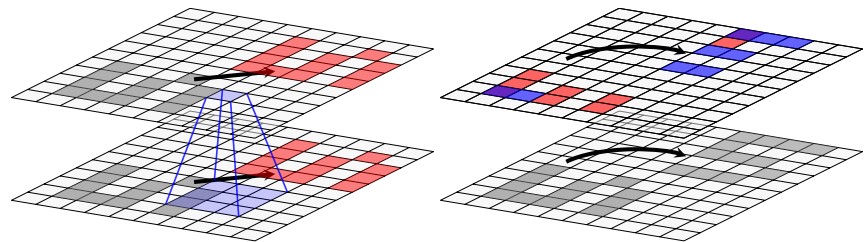

Figure 4: In convolutional neural networks if the input image is translated by some amount $(t_1, t_2)$, what used to fall in the receptive field of neuron $\mathfrak{n}_{i,j}^\ell$ is moved to the receptive field of $\mathfrak{n}_{i+t_1, j+t_2}^\ell$. Therefore, the activations transform in the very simple way $f'^\ell_{i+t_1, j+t_2} = f^\ell_{i,j}$. In contrast, rotations not only move the receptive fields around, but also permute the neurons in the receptive field internally, therefore, in general, $f'^\ell_{j,-i} \neq f^\ell_{i,j}$. The right hand figure shows that if the CNN has a horizontal filter (blue) and a vertical one (red) then their activations are exchanged by a 90 degree rotation. In steerable CNNs, if $(i,j) \mapsto (i', j')$, then $f'^\ell_{i', j'} = R(f^\ell_{i,j})$ for some fixed linear function of the rotation.

The situation in compositional networks is similar. The comp-net and message passing architectures that we have examined so far, by virtue of the aggregation function being symmetric in its arguments, are all *quasi-invariant* (with respect to permutations) in the following sense.

**Definition 5.** *Let $\mathcal{G}$ be a compound object of $n$ parts and $\mathcal{G}'$ an equivalent object in which the atoms have been permuted by some permutation $\sigma$. Let $\mathcal{N}$ be a comp-net for $\mathcal{G}$ based on an invariant composition scheme, and $\mathcal{N}'$ be the corresponding network for $\mathcal{G}'$. We say that $\mathcal{N}$ is **quasi-invariant** if for any $\mathfrak{n}_i \in \mathcal{N}$, letting $\mathfrak{n}'_j$ be the corresponding node in $\mathcal{N}'$, $f_i = f'_j$ for any $\sigma \in \mathbb{S}_n$*

Quasi-invariance in comp-nets is equivalent to the assertion that the activation $f_i$ at any given node must only depend on $\mathcal{P}_i = \{e_{j_1}, \ldots, e_{j_k}\}$ as a *set*, and not on the internal ordering of the atoms $e_{j_1}, \ldots, e_{j_k}$ making up the receptive field. At first sight this seems desirable, since it is exactly what we expect from the overall representation $\phi(\mathcal{G})$. On closer examination, however, we realize that this property is potentially problematic, since it means that $\mathfrak{n}_i$ has lost all information about which vertex in its receptive field has contributed what to the aggregate information $f_i$. In the CNN analogy, we can say that we have lost information about the *orientation* of the receptive field. In particular, if, further upstream, $f_i$ is combined with some other feature vector $f_j$ from a node with an overlapping receptive field, the aggregation process has no way of taking into account which parts of the information in $f_i$ and $f_j$ come from shared vertices and which parts do not (Figure 5).

The solution is to upgrade the $\mathcal{P}_i$ receptive fields to be *ordered sets*, and explicitly establish how $f_i$ co-varies with the internal ordering of the receptive fields. To emphasize that henceforth the $\mathcal{P}_i$ sets are ordered, we will use parentheses rather than braces to denote their content.

**Definition 6.** *Let $\mathcal{G}$, $\mathcal{G}'$, $\mathcal{N}$ and $\mathcal{N}'$ be as in Definition 5. Let $\mathfrak{n}_i$ be any node of $\mathcal{N}$ and $\mathfrak{n}_j$ the corresponding node of $\mathcal{N}'$. Assume that $\mathcal{P}_i = (e_{p_1}, \ldots, e_{p_m})$ while $\mathcal{P}'_j = (e_{q_1}, \ldots, e_{q_m})$, and let $\pi \in \mathbb{S}_m$ be the permutation that aligns the orderings of the two receptive fields, i.e., for which $e_{q_{\pi(a)}} = e_{p_a}$. We say that $\mathcal{N}$ is **covariant to permutations** if for any $\pi$, there is a corresponding function $R_\pi$ such that $f'_j = R_\pi(f_i)$.*

### 4.1 FIRST ORDER COVARIANT COMP-NETS

The form of covariance prescribed by Definition 6 is very general. To make it more specific, in line with the classical literature on steerable representations, we make the assumption that the $\{f \mapsto R_\pi(f)\}_{\pi \in \mathbb{S}_m}$ maps are *linear*, and by abuse of notation, from now on simply treat them as matrices (with $R_\pi(f) = R_\pi f$). The linearity assumption automatically implies that $\{R_\pi\}_{\pi \in \mathbb{S}_m}$ is a *representation* of $\mathbb{S}_m$ in the group theoretic sense of the word (for the definition of group representations, see the Appendix)[4].

**Proposition 3.** *If for any $\pi \in \mathbb{S}_m$, the $f \mapsto R_\pi(f)$ map appearing in Definition 6 is linear, then the corresponding $\{R_\pi\}_{\pi \in \mathbb{S}_m}$ matrices form a representation of $\mathbb{S}_m$.*

---

[4]This notion of representation must not be confused with the neural networks sense of representations of objects, as in "$f_i^\ell$ is a representation of $\mathcal{P}_i^\ell$"

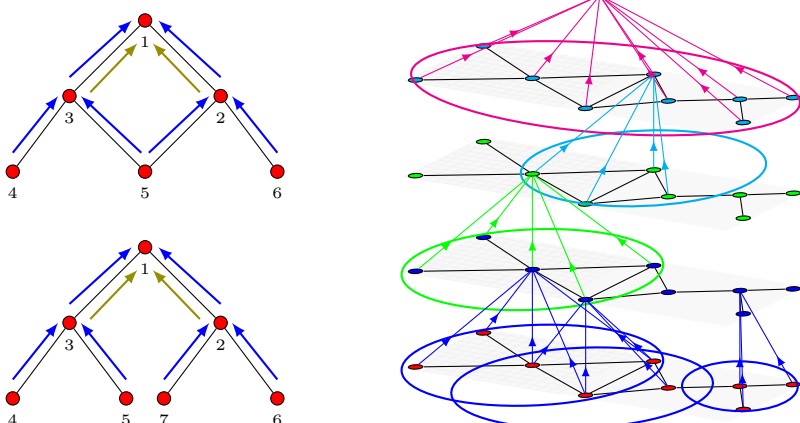

Figure 5: **Top left:** At level $\ell = 1$ $\mathfrak{n}_3$ aggregates information from $\{\mathfrak{n}_4, \mathfrak{n}_5\}$ and $\mathfrak{n}_2$ aggregates information $\{\mathfrak{n}_5, \mathfrak{n}_6\}$. At $\ell = 2$, $\mathfrak{n}_1$ collects this summary information from $\mathfrak{n}_3$ and $\mathfrak{n}_2$. **Bottom left:** This graph is not isomorphic to the top one, but the activations of $\mathfrak{n}_3$ and $\mathfrak{n}_2$ at $\ell = 1$ will be identical. Therefore, at $\ell = 2$, $\mathfrak{n}_1$ will get the same inputs from its neighbors, irrespective of whether or not $\mathfrak{n}_5$ and $\mathfrak{n}_7$ are the same node or not. **Right:** Aggregation at different levels. For keeping the figure legible only the neighborhood around one node in higher levels is marked.

The representation theory of symmetric groups is a rich subject that goes beyond the scope of the present paper (Sagan, 2001). However, there is one particular representation of $\mathbb{S}_m$ that is likely familiar even to non-algebraists, the so-called *defining representation*, given by the $P_\pi \in \mathbb{R}^{n \times n}$ permutation matrices

$$[P_\pi]_{i,j} = \begin{cases} 1 & \text{if } \pi(j) = i \\ 0 & \text{otherwise.} \end{cases}$$

It is easy to verify that $P_{\pi_2 \pi_1} = P_{\pi_2} P_{\pi_1}$ for any $\pi_1, \pi_2 \in \mathbb{S}_m$, so $\{P_\pi\}_{\pi \in \mathbb{S}_m}$ is indeed a representation of $\mathbb{S}_m$. If the transformation rules of the $f_i$ activations in a given comp-net are dictated by this representation, then each $f_i$ must necessarily be a $|\mathcal{P}_i|$ dimensional vector, and intuitively each component of $f_i$ carries information related to one specific atom in the receptive field, or the interaction of that specific atom with all the others. We call this case **first order permutation covariance**.

**Definition 7.** *We say that $\mathfrak{n}_i$ is a **first order covariant node** in a comp-net if under the permutation of its receptive field $\mathcal{P}_i$ by any $\pi \in \mathbb{S}_{|\mathcal{P}_i|}$, its activation trasforms as $f_i \mapsto P_\pi f_i$.*

## 4.2 SECOND ORDER COVARIANT COMP-NETS

It is easy to verify that given any representation $(R_g)_{g \in \mathfrak{G}}$ of a group $\mathfrak{G}$, the matrices $(R_g \otimes R_g)_{g \in \mathfrak{G}}$ also furnish a representation of $\mathfrak{G}$. Thus, one step up in the hierarchy from $P_\pi$–covariant comp-nets are $P_\pi \otimes P_\pi$–covariant comp-nets, where the $f_i$ feature vectors are now $|\mathcal{P}_i|^2$ dimensional vectors that transform under permutations of the internal ordering by $\pi$ as $f_i \mapsto (P_\pi \otimes P_\pi) f_i$.

If we reshape $f_i$ into a matrix $F_i \in \mathbb{R}^{|\mathcal{P}_i| \times |\mathcal{P}_i|}$, then the action

$$F_i \mapsto P_\pi F_i P_\pi^\top$$

is equivalent to $P_\pi \otimes P_\pi$ acting on $f_i$. In the following, we will prefer this more intuitive matrix view, since it clearly expresses that feature vectors that transform this way express *relationships* between the different constituents of the receptive field. Note, in particular, that if we define $A\downarrow_{\mathcal{P}_i}$ as the restriction of the adjacency matrix to $\mathcal{P}_i$ (i.e., if $\mathcal{P}_i = (e_{p_1}, \ldots, e_{p_m})$ then $[A\downarrow_{\mathcal{P}_i}]_{a,b} = A_{p_a, p_b}$), then $A\downarrow_{\mathcal{P}_i}$ transforms exactly as $F_i$ does in the equation above.

**Definition 8.** *We say that $\mathfrak{n}_i$ is a **second order covariant node** in a comp-net if under the permutation of its receptive field $\mathcal{P}_i$ by any $\pi \in \mathbb{S}_{|\mathcal{P}_i|}$, its activation transforms as $F_i \mapsto P_\pi F_i P_\pi^\top$.*

### 4.3 THIRD AND HIGHER ORDER COVARIANT COMP-NETS

Taking the pattern further lets us consider third, fourth, and general, $k$'th order nodes in our comp-net, in which the activations are $k$'th order tensors, transforming under permutations as

$$F_i \mapsto F_i' \qquad \text{where} \qquad [F_i']_{j_1,\ldots,j_k} = \sum_{j_1'} \sum_{j_2'} \cdots \sum_{j_k'} [P_\pi]_{j_1,j_1'} [P_\pi]_{j_2,j_2'} \cdots [P_\pi]_{j_k,j_k'} [F_i]_{j_1',\ldots,j_k'},$$

In the more compact, so called Einstein notation[5],

$$[F_i']_{j_1,\ldots,j_k} = [P_\pi]_{j_1}^{j_1'} [P_\pi]_{j_2}^{j_2'} \cdots [P_\pi]_{j_k}^{j_k'} [F_i]_{j_1',\ldots,j_k'}. \tag{1}$$

In general, we will call any quantity which transforms according to this equation a **$k$'th order P-tensor**. Note that this notion of tensors is distinct from the common usage of the term in neural networks, and more similar to how the word is used in Physics, because it not only implies that $F_i$ is a quanity representable by an $m \times m \times \ldots \times m$ array of numbers, but also that $F_i$ transforms in a specific way.

Since scalars, vectors and matrices can be considered as $0^{\text{th}}$, $1^{\text{st}}$ and $2^{\text{nd}}$ order tensors, respectively, the following definition covers Definitions 5, 7 and 8 as special cases (with quasi-invariance being equivalent to zeroth order equivariance). To unify notation and terminology, regardless of the dimensionality, in the following we will always talk about *feature tensors* rather than *feature vectors*, and denote the activations with $F_i$ rather than $f_i$, as we did in the first half of the paper.

**Definition 9.** *We say that $\mathfrak{n}_i$ is a **$k$'th order covariant node** in a comp-net if the corresponding activation $F_i$ is a $k$'th order P–tensor, i.e., it transforms under permutations of $\mathcal{P}_i$ according to (1), or the activation is a sequence of $c$ separate P–tensors $F_i^{(1)}, \ldots, F_i^{(c)}$ corresponding to $c$ distinct channels.*

## 5 TENSOR AGGREGATION RULES

The previous sections prescribed how activations must transform in comp-nets of different orders, but did not explain how this can be assured, and what it entails for the $\Phi$ aggregation functions. Fortunately, tensor arithmetic provides a compact framework for deriving the general form of these operations. Recall the four basic operations that can be applied to tensors[6]:

1. The **tensor product** of $A \in \mathcal{T}^k$ with $B \in \mathcal{T}^p$ yields a tensor $C = A \otimes B \in \mathcal{T}^{p+k}$ where

$$C_{i_1,i_2,\ldots,i_{k+p}} = A_{i_1,i_2,\ldots,i_k} B_{i_{k+1},i_{k+2},\ldots,i_{k+p}}.$$

2. The **elementwise product** of $A \in \mathcal{T}^k$ with $B \in \mathcal{T}^p$ along dimensions $(a_1, a_2, \ldots, a_p)$ yields a tensor $C = A \odot_{(a_1,\ldots,a_p)} B \in \mathcal{T}^k$ where

$$C_{i_1,i_2,\ldots,i_k} = A_{i_1,i_2,\ldots,i_k} B_{i_{a_1},i_{a_2},\ldots,i_{a_p}}.$$

3. The **projection (summation)** of $A \in \mathcal{T}^k$ along dimensions $\{a_1, a_2, \ldots, a_p\}$ yields a tensor $C = A\!\downarrow_{a_1,\ldots,a_p} \in \mathcal{T}^{k-p}$ with

$$C_{i_1,i_2,\ldots,i_k} = \sum_{i_{a_1}} \sum_{i_{a_2}} \cdots \sum_{i_{a_p}} A_{i_1,i_2,\ldots,i_k},$$

where we assume that $i_{a_1}, \ldots, i_{a_p}$ have been removed from amongst the indices of $C$.

4. The **contraction** of $A \in \mathcal{T}^k$ along the pair of dimensions $\{a, b\}$ (assuming $a < b$) yields a $k-2$ order tensor

$$C_{i_1,i_2,\ldots,i_k} = \sum_j A_{i_1,\ldots,i_{a-1},j,i_{a+i},\ldots,i_{b-1},j,i_{b+1},\ldots,k},$$

---

[5]The Einstein convention is that if, in a given tensor expression the same index appears twice, once "upstairs" and once "downstairs", then it is summed over. For example, the matrix/vector product $y = Ax$ would be written $y_i = A_i^j x_j$

[6]Here and in the following $\mathcal{T}^k$ will denote the class of $k$'th order tensors ($k$ dimensional tensors), regardless of their transformation properties.

where again we assume that $i_a$ and $i_b$ have been removed from amongst the indices of $C$. Using Einstein notation this can be written much more compactly as

$$C_{i_1,i_2,\ldots,i_k} = A_{i_1,i_2,\ldots,i_k} \delta^{i_a,i_b},$$

where $\delta^{i_a,i_b}$ is the diagonal tensor with $\delta^{i,j} = 1$ if $i = j$ and 0 otherwise. In a somewhat unorthodox fashion, we also generalize contractions to (combinations of) larger sets of indices $\{\{a_1^1, \ldots, a_{p_1}^1\}, \{a_1^2, \ldots, a_{p_2}^2\}, \ldots, \{a_1^q, \ldots, a_{p_q}^q\}\}$ as the $(k - \sum_j p_j)$ order tensor

$$C_{\ldots} = A_{i_1,i_2,\ldots,i_k} \, \delta^{a_1^1,\ldots,a_{p_1}^1} \, \delta^{a_1^2,\ldots,a_{p_2}^2} \, \ldots \, \delta^{a_1^q,\ldots,a_{p_q}^q}.$$

Note that this subsumes projections, since it allows us to write $A\!\downarrow_{a_1,\ldots,a_p}$ in the slightly unusual looking form

$$A\!\downarrow_{a_1,\ldots,a_p} = A_{i_1,i_2,\ldots,i_k} \, \delta^{i_{a_1}} \delta^{i_{a_2}} \, \ldots \, \delta^{i_{a_k}}.$$

The following proposition shows that, remarkably, all of the above operations (as well as taking linear conbinations) preserve the way that $P$–tensors behave under permutations and thus they can be freely "mixed and matched" within $\Phi$.

**Proposition 4.** *Assume that $A$ and $B$ are $k$'th and $p$'th order $P$–tensors, respectively. Then*

1. *$A \otimes B$ is a $k+p$'th order $P$–tensor.*
2. *$A \odot_{(a_1,\ldots,a_p)} B$ is a $k$'th order $P$–tensor.*
3. *$A\!\downarrow_{a_1,\ldots,a_p}$ is a $k-p$'th order $P$–tensor.*
4. *$A_{i_1,i_2,\ldots,i_k} \, \delta^{a_1^1,\ldots,a_{p_1}^1} \ldots \delta^{a_1^q,\ldots,a_{p_q}^q}$ is a $k - \sum_j p_j$'th order $P$–tensor.*

*In addition, if $A_1, \ldots, A_u$ are $k$'th order $P$–tensors and $\alpha_1, \ldots, \alpha_u$ are scalars, then $\sum_j \alpha_j A_j$ is a $k$'th order $P$–tensor.*

The more challenging part of constructing the aggregation scheme for comp-nets is establishing how to relate $P$–tensors at different nodes. The following two propositions answer this question.

**Proposition 5.** *Assume that node $\mathfrak{n}_a$ is a descendant of node $\mathfrak{n}_b$ in a comp-net $\mathcal{N}$, $\mathcal{P}_a = (e_{p_1}, \ldots, e_{p_m})$ and $\mathcal{P}_b = (e_{q_1}, \ldots, e_{q_{m'}})$ are the corresponding ordered receptive fields (note that this implies that, as sets, $\mathcal{P}_a \subseteq \mathcal{P}_b$), and $\chi^{a \to b} \in \mathbb{R}^{m \times m'}$ is an indicator matrix defined*

$$\chi_{i,j}^{a \to b} = \begin{cases} 1 & \text{if } q_j = p_i \\ 0 & \text{otherwise.} \end{cases}$$

*Assume that $F$ is a $k$'th order $P$–tensor with respect to permutations of $(e_{p_1}, \ldots, e_{p_m})$. Then, dropping the $^{a \to b}$ superscript for clarity,*

$$\widetilde{F}_{i_1,\ldots,i_k} = \chi_{i_1}^{j_1} \chi_{i_2}^{j_2} \, \cdots \, \chi_{i_k}^{j_k} \, F_{j_1,\ldots,j_k} \tag{2}$$

*is a $k$'th order $P$–tensor with respect to permutations of $(e_{q_1}, \ldots, e_{q_{m'}})$.*

Equation 2 tells us that when node $\mathfrak{n}_b$ aggregates $P$–tensors from its children, it first has to "promote" them to being $P$–tensors with respect to the contents of its own receptive field by contracting along each of their dimensions with the appropriate $\chi^{a \to b}$ matrix. This is a critical element in comp-nets to guarantee covariance.

**Proposition 6.** *Let $\mathfrak{n}_{c_1}, \ldots, \mathfrak{n}_{c_s}$ be the children of $\mathfrak{n}_t$ in a message passing type comp-net with corresponding $k$'th order tensor activations $F_{c_1}, \ldots, F_{c_s}$. Let*

$$[\widetilde{F}_{c_u}]_{i_1,\ldots,i_k} = [\chi^{c_u \to t}]_{i_1}^{j_1} [\chi^{c_u \to t}]_{i_2}^{j_2} \, \cdots \, [\chi^{c_u \to t}]_{i_k}^{j_k} [F_{c_u}]_{j_1,\ldots,j_k}$$

*be the promotions of these activations to $P$–tensors of $\mathfrak{n}_t$. Assume that $\mathcal{P}_t = (e_{p_1}, \ldots, e_{p_m})$. Now let $\overline{F}$ be a $k+1$'th order object in which the $j$'th slice is $\widetilde{F}_{p_j}$ if $\mathfrak{n}_{p_j}$ is one of the children of $\mathfrak{n}_t$, i.e.,*

$$\overline{F}_{i_1,\ldots,i_k,j} = [\widetilde{F}_{p_j}]_{i_1,\ldots,i_k},$$

*and zero otherwise. Then $\overline{F}$ is a $k+1$'th order $P$–tensor of $\mathfrak{n}_t$.*

Finally, as already mentioned, the restriction of the adjacency matrix to $\mathcal{P}_i$ is a second order $P$–tensor, which gives an easy way of explicitly adding topological information to the activation.

**Proposition 7.** *If $F_i$ is a $k$'th order $P$–tensor at node $\mathfrak{n}_i$, and $A\!\downarrow_{\mathcal{P}_i}$ is the restriction of the adjacency matrix to $\mathcal{P}_i$ as defined in Section 4.2, then $F \otimes A\!\downarrow_{\mathcal{P}_i}$ is a $k+2$'th order $P$–tensor.*

## 5.1 THE GENERAL AGGREGATION FUNCTION AND ITS SPECIAL CASES

Combining all the above results, assuming that node $\mathfrak{n}_t$ has children $\mathfrak{n}_{c_1}, \ldots, \mathfrak{n}_{c_s}$, we arrive at the following general algorithm for the aggregation rule $\Phi_t$:

---

1. Collect all the $k$'th order activations $F_{c_1}, \ldots, F_{c_s}$ of the children.
2. Promote each activation to $\widetilde{F}_{c_1}, \ldots, \widetilde{F}_{c_s}$ (Proposition 5).
3. Stack $\widetilde{F}_{c_1}, \ldots, \widetilde{F}_{c_s}$ together into a $k+1$ order tensor $T$ (Proposition 6).
4. Optionally form the tensor product of $T$ with $A{\downarrow}_{\mathcal{P}_t}$ to get a $k+3$ order tensor $H$ (otherwise just set $H = T$) (Proposition 7).
5. Contract $H$ along some number of combinations of dimensions to get $s$ separate lower order tensors $Q_1, \ldots, Q_s$ (Proposition 4).
6. Mix $Q_1, \ldots, Q_s$ with a matrix $W \in \mathbb{R}^{s' \times s}$ and apply a nonlinearity $\Upsilon$ to get the final activation of the neuron, which consists of the $s'$ output tensors

$$F^{(i)} = \Upsilon \left[ \sum_{j=1}^{s} W_{i,j} \, Q_j + b_i \mathbb{1} \right] \qquad i = 1, 2, \ldots s',$$

where the $b_i$ scalars are bias terms, and $\mathbb{1}$ is the $|\mathcal{P}_t| \times \ldots \times |\mathcal{P}_t|$ dimensional all ones tensor.

---

A few remarks are in order about this general scheme:

1. Since $\widetilde{F}_{c_1}, \ldots, \widetilde{F}_{c_s}$ are stacked into a larger tensor and then possibly also multiplied by $A{\downarrow}_{\mathcal{P}_t}$, the general tendency would be for the tensor order to increase at every node, and the corresponding storage requirements to increase exponentially. The purpose of the contractions in Step 5 is to counteract this tendency, and pull the order of the tensors back to some small number, typically $1, 2$ or $3$.
2. However, since contractions can be done in many different ways, the number of channels will increase. When the number of input channels is small, this is reasonable, since otherwise the number of learnable weights in the algorithm would be too small. However, if unchecked, this can also become problematic. Fortunately, mixing the channels by $W$ on Step 6 gives an opportunity to stabilize the number of channels at some value $s'$.
3. In the pseudocode above, for simplicity, the number of input channels is one and the number of output channels is $s'$. More realistically, the inputs would also have multiple channels (say, $s_0$) which would be propagated through the algorithm independently up to the mixing stage, making $W$ an $s' \times s \times s_0$ dimension tensor (not in the $P$–tensor sense!).
4. The conventional part of the entire algorithm is Step 6, and the only learnable parameters are the entries of the $W$ matrix (tensor) and the $b_i$ bias terms. These parameters are shared by all nodes in the network and learned in the usual way, by stochastic gradient descent.
5. Our scheme could be elaborated further while maintaining permutation covariance by, for example taking the tensor product of $T$ with itself, or by introducing $A{\downarrow}_{\mathcal{P}_t}$ in a different way. However, the way that $\widetilde{F}_{c_1}, \ldots, \widetilde{F}_{c_s}$ and $A{\downarrow}_{\mathcal{P}_t}$ are combined by tensor products is already much more general and expressive than conventional message passing networks.
6. Our framework admits many design choices, including the choice of the order odf the activations, the choice of contractions, and $c'$. However, the overall structure of Steps 1–5 is fully dictated by the covariance constraint on the network.
7. The final output of the network $\phi(G) = F_r$ must be permutation invariant. That means that the root node $\mathfrak{n}_r$ must produce a tuple of zeroth order tensors (scalars) $(F_r^{(1)}, \ldots, F_r^{(c)})$. This is similar to how many other graph representation algorithms compute $\phi(G)$ by summing the activations at level $L$ or creating histogram features.

We consider a few special cases to explain how tensor aggregation relates to more conventional message passing rules.

### 5.1.1 Zeroth order tensor aggregation

Constraining both the input tensors $F_{c_1}, \dots, F_{c_s}$ and the outputs to be zeroth order tensors, i.e., scalars, and foregoing multiplication by $A\!\downarrow_{\mathcal{P}_t}$ greatly simplifies the form of $\Phi$. In this case there is no need for promotions, and $T$ is just the vector $(F_{c_1}^\ell, \dots, F_{c_s}^\ell)$. There is only one way to contract a vector into a scalar, and that is to sum its elements. Therefore, in this case, the entire aggregation algorithm reduces to the simple formula

$$F_i = \Upsilon\Big( w \sum_{u=1}^{c} F_{c_u} + b \Big).$$

For a neural network this is too simplistic. However, it's interesting to note that the Weisfeiler–Lehmann isomorphism test essentially builds on just this formula, with a specific choice of $\Upsilon$ (Read & Corneil, 1977). If we allow more channels in the inputs and the outputs, $W$ becomes a matrix, and we recover the simplest form of neural message passing algorithms (Duvenaud et al., 2015).

### 5.1.2 First order tensor aggregation

In first order tensor aggregation, assuming that $|\mathcal{P}_i| = m$, $\widetilde{F}_{c_1}, \dots, \widetilde{F}_{c_s}$ are $m$ dimensional column vectors, and $T$ is an $m \times m$ matrix consisting of $\widetilde{F}_{c_1}, \dots, \widetilde{F}_{c_s}$ stacked columnwise. There are two ways of contracting (in our generalized sense) a matrix into a vector: by summing over its rows, or summing over its columns. The second of these choices leads us back to summing over all contributions from the children, while the first is more interesting because it corresponds to summing $\widetilde{F}_{c_1}, \dots, \widetilde{F}_{c_s}$ as vectors individually. In summary, we get an aggregation function that transforms a single input channel to two output channels of the form

$$F_i^{(1)} = \Upsilon\Big[ w_{1,1}(T^\top \mathbf{1}) + w_{1,2}(T\,\mathbf{1}) + b_1\,\mathbf{1} \Big], \qquad F_i^{(2)} = \Upsilon\Big[ w_{2,1}(T^\top \mathbf{1}) + w_{2,2}(T\,\mathbf{1}) + b_2\,\mathbf{1} \Big],$$

where $\mathbf{1}$ denotes the $m$ dimensional all ones vector. Thus, in this layer $W \in \mathbb{R}^{2\times 2}$. Unless constrained by $c'$, in each subsequent layer the number of channels doubles further and these channels can all mix with each other, so $W^{(2)} \in \mathbb{R}^{4\times 4}$, $W^{(3)} \in \mathbb{R}^{8\times 8}$, and so on.

### 5.1.3 Second order tensor aggregation without the adjacency matrix

In second order tensor aggregation, $T$ is a third order $P$–tensor, which can be contracted back to second order in three different ways, by projecting it along each of its dimensions. Therefore the outputs will be the three matrices

$$F^{(i)} = \Upsilon\big(w_{i,1}T\!\downarrow_1 + w_{i,2}T\!\downarrow_2 + w_{i,3}T\!\downarrow_3 + b_i\mathbf{1}_{m\times m}\big) \qquad\qquad i \in \{1,2,3\},$$

and the weight matrix is $W \in \mathbb{R}^{3\times 3}$.

### 5.1.4 Second order tensor aggregation with the adjacency matrix

The first nontrivial tensor contraction case occurs when $\widetilde{F}_{c_1}, \dots, \widetilde{F}_{c_s}$ are second order tensors, and we multiply with $A\!\downarrow_{\mathcal{P}_t}$, since in that case $T$ is 5th order, and can be contracted down to second order in a total of 50 different ways:

1. The "1+1+1" case contracts $T$ in the form $T_{i_1,i_2,i_3,i_4,i_5}\delta^{i_{a_1}}\delta^{i_{a_2}}\delta^{i_{a_3}}$, i.e., it projects $T$ down along 3 of its 5 dimensions. This alone can be done in $\binom{5}{3} = 10$ different ways[7]
2. The "1+2" case contracts $T$ in the form $T_{i_1,i_2,i_3,i_4,i_5}\delta^{i_{a_1}}\delta^{i_{a_2},i_{a_3}}$, i.e., it projects $T$ along one dimension, and contracts it along two others. This can be done in $3\binom{5}{3} = 30$ ways.
3. The "3" case is a single 3-fold contraction $T_{i_1,i_2,i_3,i_4,i_5}\delta^{i_{a_1},i_{a_2},i_{a_3}}$, which again can be done in $\binom{5}{3} = 10$ different ways.

The tensor $\mathcal{T}_{i_1,i_2,i_3,i_4,i_5}$ will be symmetric with respect to two sets of indices, following the structure of the promotion tensors and the adjacency matrix. Including these symmetries, the number of contractions is 18 including: five "1+1+1", ten "1+2", and three "3".

---

[7]For simplicity, we ignore the fact that symmetries, such as the symmetry of $A\!\downarrow_{\mathcal{P}_t}$, might reduce the number of distinct projections somewhat.

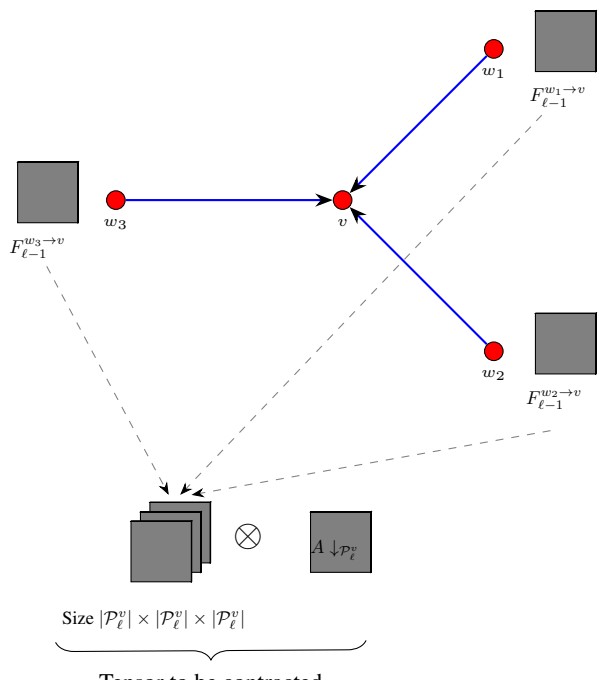

Figure 6: The activations of vertices in the receptive field $\mathcal{P}_\ell^v = \{w_1, w_2, w_3\}$ of vertex $v$ at level $\ell$-th are stacked into a 3rd order tensor and undergo a tensor product operation with the restricted adjacency matrix, and then contracted in different ways. In this figure, we only consider single channel, each channel is represented by a 5th order tensor. In the general case of multi channels, the resulting tensor would have 6th order, but we contract on each channel separately.

## 6 EXPERIMENTS

We compared the second order variant (CCN 2D) of our CCNs framework (Section 4.2) to several standard graph learning algorithms on three types of datasets that involve learning the properties of molecules from their structure:

1. **The Harvard Clean Energy Project** (Hachmann et al., 2011), consisting of 2.3 million organic compounds that are candidates for use in solar cells. The regression target in this case is Power Conversion Efficiency (PCE). Due to time constraints, instead of using the entire dataset, the experiments were ran on a random subset of 50,000 molecules.

2. **QM9**, which is a dataset of all 133k organic molecules with up to nine heavy atoms (C,O,N and F) out of the GDB-17 universe of molecules. Each molecule has 13 target properties to predict. The dataset does contain spatial information relating to the atomic configurations, but we only used the chemical graph and atom node labels. For our experiments we normalized each target variable to have mean 0 and standard deviation 1. We report both MAE and RMSE for all normalized learning targets.

3. **Graph kernels datasets**, specifically (a) MUTAG, which is a dataset of 188 mutagenic aromatic and heteroaromatic compounds (Debnat et al., 1991); (b) PTC, which consists of 344 chemical compounds that have been tested for positive or negative toxicity in lab rats (Toivonen et al., 2003); (c) NCI1 and NCI109, which have 4110 and 4127 compounds respectively, each screened for activity against small cell lung cancer and ovarian cancer lines (Wale et al., 2008).

In the case of HCEP, we compared CCN to lasso, ridge regression, random forests, gradient boosted trees, optimal assignment Wesifeiler–Lehman graph kernel (Kriege et al., 2016) (WL), neural graph fingerprints (Duvenaud et al., 2015), and the "patchy-SAN" convolutional type algorithm from (Niepert et al., 2016) (referred to as PSCN). For the first four of these baseline methods, we created simple feature vectors from each molecule: the number of bonds of each type (i.e. number of

H–H bonds, number of C–O bonds, etc) and the number of atoms of each type. Molecular graph fingerprints uses atom labels of each vertex as base features. For ridge regression and lasso, we cross validated over $\lambda$. For random forests and gradient boosted trees, we used 400 trees, and cross validated over max depth, minimum samples for a leaf, minimum samples to split a node, and learning rate (for GBT). For neural graph fingerprints, we used 2 layers and a hidden layer size of 10. In PSCN, we used a patch size of 10 with two convolutional layers and a dense layer on top as described in their paper.

For the graph kernels datasets, we compare against graph kernel results as reported in (Kondor & Pan, 2016) (which computed kernel matrices using the Weisfeiler–Lehman, Weisfeiler–edge, shortest paths, graphlets and multiscale Laplacian graph kernels and used a C-SVM on top), Neural graph fingerprints (with 2 levels and a hidden size of 10) and PSCN. For QM9, we compared against the Weisfeiler–Lehman graph kernel (with C-SVM on top), neural graph fingerprints, and PSCN. The settings for NGF and PSCN are as described for HCEP.

For our own method, second order CCN, we initialized the base features of each vertex with computed histogram alignment features, inspired by (Kriege et al., 2016), of depth up to 10. Each vertex receives a base label $l_i = \mathrm{concat}_{j=1}^{10} H_j(i)$ where $H_j(i) \in \mathbb{R}^d$ (with $d$ being the total number of distinct discrete node labels) is the vector of relative frequencies of each label for the set of vertices at distance equal to $j$ from vertex $i$. We use exactly 18 unique contractions defined in 5.1.4 that result in additional channels. We used up to three levels and the intermediate number of channels increases 18 time at each level. To avoid exponentially growing channels, we applied learnable weight matrices to compress the channels into a fixed number of channels.

In each experiment we used 80% of the dataset for training, 10% for validation, and evaluated on the remaining 10% test set. For the kernel datasets we performed the experiments on 10 separate training/validation/test stratified splits and averaged the resulting classification accuracies. We used Adam optimization method (Kingma & Ba, 2015). Our initial learning rate was set to 0.001 after experimenting on a held out set. The learning rate decayed linearly after each step towards a minimum of $10^{-6}$.

## 6.1 GraphFlow Deep Learning Framework

We developed our custom Deep Learning framework in C++/CUDA named GraphFlow that supports symbolic/automatic differentiation, dynamic computation graphs, specialized tensor operations, and computational acceleration with GPU. Our method, Covariant Compositional Networks, and other graph neural networks such as Neural Graph Fingerprints (Duvenaud et al., 2015), PSCN (Niepert et al., 2016) and Gated Graph Neural Networks (Li et al., 2016) are implemented based on the Graph-Flow framework. Our source code can be found at `https://github.com/HyTruongSon/GraphFlow`.

One challenge of the implementation of Covariant Compositional Networks is that the high-order tensors (for example, in figure 6, we have a 5th order tensor after the tensor product operation) cannot be stored explicitly in the memory. Our solution is to propose a *virtual indexing system* in such a way that we never compute the whole sparse high-order tensor at once, but only compute its elements when given the indices. Basically, we always work with a *virtual tensor*, and that allows us to implement our tensor reduction/contraction operations efficiently with GPU.

## 6.2 Discussion

On the subsampled HCEP dataset, CCN outperforms all other methods by a very large margin. For the graph kernels datasets, SVM with the Weisfeiler–Lehman kernels achieve the highest accuracy on NCI1 and NCI109, while CCN wins on MUTAG and PTC. Perhaps this poor performance is to be expected, since the datasets are small and neural network approaches usually require tens of thousands of training examples at minimum to be effective. Indeed, neural graph fingerprints and PSCN also perform poorly compared to the Weisfeiler–Lehman kernels.

In the QM9 experiments, CCN beats the three other algorithms in both mean absolute error and root mean squared error. It should be noted that (Gilmer et al., 2017) obtained stronger results on QM9, but we cannot properly compare our results with theirs because our experiments only use

Table 1: HCEP regression results

|  | Test MAE | Test RMSE |
|---|---|---|
| Lasso | 0.867 | 1.437 |
| Ridge regression | 0.854 | 1.376 |
| Random forest | 1.004 | 1.799 |
| Gradient boosted trees | 0.704 | 1.005 |
| WL graph kernel | 0.805 | 1.096 |
| Neural graph fingerprints | 0.851 | 1.177 |
| PSCN ($k = 10$) | 0.718 | 0.973 |
| CCN 2D | **0.340** | **0.449** |

Table 2: Kernel Datasets Classification results (accuracy +/- standard deviation)

|  | MUTAG | PTC | NCI1 | NCI109 |
|---|---|---|---|---|
| WL | $84.50 \pm 2.16$ | $59.97 \pm 1.60$ | **$84.76 \pm 0.32$** | **$85.12 \pm 0.29$** |
| WL-edge | $82.94 \pm 2.33$ | $60.18 \pm 2.19$ | **$84.65 \pm 0.25$** | **$85.32 \pm 0.34$** |
| SP | $85.50 \pm 2.50$ | $59.53 \pm 1.71$ | $73.61 \pm 0.36$ | $73.23 \pm 0.26$ |
| Graphlet | $82.44 \pm 1.29$ | $55.88 \pm 0.31$ | $62.40 \pm 0.27$ | $62.35 \pm 0.28$ |
| p-RW | $80.33 \pm 1.35$ | $59.85 \pm 0.95$ | TIMED OUT | TIMED OUT |
| MLG | $87.94 \pm 1.61$ | $63.26 \pm 1.48$ | $81.75 \pm 0.24$ | $81.31 \pm 0.22$ |
| PSCN $k = 10$ (Niepert et al.) | $88.95 \pm 4.37$ | $62.29 \pm 5.68$ | $76.34 \pm 1.68$ | N/A |
| Neural graph fingerprints | $89.00 \pm 7.00$ | $57.85 \pm 3.36$ | $62.21 \pm 4.72$ | $56.11 \pm 4.31$ |
| CCN 2D | **$91.64 \pm 7.24$** | **$70.62 \pm 7.04$** | $76.27 \pm 4.13$ | $75.54 \pm 3.36$ |

the adjacency matrices and atom labels of each node, while theirs includes comprehensive chemical features that better inform the target quantum properties.

# 7 CONCLUSIONS

We have presented a general framework called covariant compositional networks (CCNs) for constructing covariant graph neural networks, which encompasses other message passing approaches as special cases, but takes a more general and principled approach to ensuring covariance with respect to permutations. Experimental results on several benchmark datasets show that CCNs can outperform other state-of-the-art algorithms.

Table 3: QM9 regression results (MAE)

|  | WLGK | NGF | PSCN ($k = 10$) | CCN 2D |
|---|---|---|---|---|
| alpha | 0.46 | 0.43 | 0.20 | **0.16** |
| Cv | 0.59 | 0.47 | 0.27 | **0.23** |
| G | 0.51 | 0.46 | 0.33 | **0.29** |
| gap | 0.72 | 0.67 | 0.60 | **0.54** |
| H | 0.52 | 0.47 | 0.34 | **0.30** |
| HOMO | 0.64 | 0.58 | 0.51 | **0.39** |
| LUMO | 0.70 | 0.65 | 0.59 | **0.53** |
| mu | 0.69 | 0.63 | 0.54 | **0.48** |
| omega1 | 0.72 | 0.63 | 0.57 | **0.45** |
| R2 | 0.55 | 0.49 | 0.22 | **0.19** |
| U | 0.52 | 0.47 | 0.34 | **0.29** |
| U0 | 0.52 | 0.47 | 0.34 | **0.29** |
| ZPVE | 0.57 | 0.51 | 0.43 | **0.39** |

Table 4: QM9 regression results (RMSE)

|        | WLGK | NGF  | PSCN ($k = 10$) | CCN 2D |
|--------|------|------|------|------|
| alpha  | 0.68 | 0.65 | 0.31 | **0.26** |
| Cv     | 0.78 | 0.65 | 0.34 | **0.30** |
| G      | 0.67 | 0.62 | 0.43 | **0.38** |
| gap    | 0.86 | 0.82 | 0.75 | **0.69** |
| H      | 0.68 | 0.62 | 0.44 | **0.40** |
| HOMO   | 0.91 | 0.81 | 0.70 | **0.55** |
| LUMO   | 0.84 | 0.79 | 0.73 | **0.68** |
| mu     | 0.92 | 0.87 | 0.75 | **0.67** |
| omega1 | 0.84 | 0.77 | 0.73 | **0.65** |
| R2     | 0.81 | 0.71 | 0.31 | **0.27** |
| U      | 0.67 | 0.62 | 0.44 | **0.40** |
| U0     | 0.67 | 0.62 | 0.44 | **0.39** |
| ZPVE   | 0.72 | 0.66 | 0.55 | **0.51** |

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

## A  MATHEMATICAL BACKGROUND

**Groups.**   A **group** is a set $G$ endowed with an operation $G \times G \to G$ (usually denoted multiplicatively) obeying the following axioms:

G1.  for any $u, v \in G$, $uv \in G$ (closure);
G2.  for any $u, v, w \in G$, $u(vw) = (uv)w$ (associativity);
G3.  there is a unique $e \in G$, called the **identity** of $G$, such that $eu = ue = u$ for any $u \in G$;
G4.  for any $u \in G$, there is a corresponding element $u^{-1} \in G$ called the **inverse** of $u$, such that $uu^{-1} = u^{-1}u = e$.

We do *not* require that the group operation be commutative, i.e., in general, $uv \neq vu$. Groups can be finite or infinite, countable or uncountable, compact or non-compact. While most of the results in this paper would generalize to any compact group, the keep the exposition as simple as possible, throughout we assume that $G$ is finite or countably infinite. As usual, $|G|$ will denote the size (cardinality) of $G$, sometimes also called the **order** of the group.

**Representations.**   A (finite dimensional) **representation** of a group $G$ over a field $\mathbb{F}$ is a matrix-valued function $R \colon G \to \mathbb{F}^{d_\rho \times d_\rho}$ such that $R(x)R(y) = R(xy)$ for any $x, y \in G$. We generally assume that $\mathbb{F} = \mathbb{C}$, however in the special case when $G$ is the symmetric group $\mathbb{S}_n$ we can restrict ourselves to only considering real-valued representations, i.e., $\mathbb{F} = \mathbb{R}$.

## B  PROOFS

**Proof of Proposition 1.** Let $\mathcal{G}$ and $\mathcal{G}'$ be two compound objects, where $\mathcal{G}'$ is equivalent to $\mathcal{G}$ up to a permutation $\sigma \in \mathbb{S}_n$ of the atoms. For any node $\mathfrak{n}_a$ of $\mathcal{G}$ we let $\mathfrak{n}'_a$ be the corresponding node of $\mathcal{G}'$, and let $f_a$ and $f'_a$ be their activations.

We prove that $f_a = f'_a$ for every node in $\mathcal{G}$ by using induction on the distance of $\mathfrak{n}_a$ from its farthest descendant that is a leaf, which we call its *height* and denote $h(a)$. For $h(a) = 0$, the statment is

clearly true, since $f_a = f'_a = \ell_{\xi(a)}$. Now assume that it is true for all nodes with height up to $h^*$. For any node $\mathfrak{n}_a$ with $h(a) = h^* + 1$, $f_a = \Phi(f_{c_1}, f_{c_2}, \ldots, f_{c_k})$, where each of the children $c_1, \ldots, c_k$ are of height at most $h^*$, therefore

$$f_a = \Phi(f_{c_1}, f_{c_2}, \ldots, f_{c_k}) = \Phi(f'_{c_1}, f'_{c_2}, \ldots, f'_{c_k}) = f'_a.$$

Thus, $f_a = f'_a$ for every node in $\mathcal{G}$. The proposition follows by $\phi(\mathcal{G}) = f_r = f'_r = \phi(\mathcal{G}')$. ∎

**Proof of Proposition 3.** Let $\mathcal{G}$, $\mathcal{G}'$, $\mathcal{N}$ and $\mathcal{N}'$ be as in Definition 5. As in Definition 6, for each node (neuron) $\mathfrak{n}_i$ in $\mathcal{N}$ there is a node $\mathfrak{n}'_j$ in $\mathcal{N}'$ such that their receptive fields are equivalent up to permutation. That is, if $|\mathcal{P}_i| = m$, then $|\mathcal{P}'_j| = m$, and there is a permutation $\pi \in \mathbb{S}_m$, such that if $\mathcal{P}_i = (e_{p_1}, \ldots, e_{p_m})$ and $\mathcal{P}'_j = (e_{q_1}, \ldots, e_{q_m})$, then $e_{q_{\pi(a)}} = e_{p_a}$. By covariance, then $f'_j = R_\pi(f_i)$.

Now let $\mathcal{G}''$ be a third equivalent object, and $\mathcal{N}''$ the corresponding comp-net. $\mathcal{N}''$ must also have a node, $\mathfrak{n}''_k$, that corresponds to $\mathfrak{n}_i$ and $\mathfrak{n}'_j$. In particular, letting its receptive field be $\mathcal{P}''_k = (e_{r_1}, \ldots, e_{r_m})$, there is a permutation $\sigma \in \mathbb{S}_m$ for which $e_{r_{\sigma(b)}} = e_{q_b}$. Therefore, $f''_k = R_\sigma(f'_j)$.

At the same time, $\mathfrak{n}''_k$ is also in correspondence with $\mathfrak{n}_i$. In particular, letting $\tau = \sigma\pi$ (which corresponds to first applying the permutation $\pi$, then applying $\sigma$), $e_{r_{\tau(a)}} = e_{p_a}$, and therefore $f''_k = R_\tau(f_i)$. Hence, the $\{R_\pi\}$ maps must satisfy

$$R_{\sigma\pi}(f_i) = R_\sigma(f'_j) = R_\sigma(R_\pi(f_i)),$$

for any $f_i$. More succinctly, $R_{\sigma\pi} = R_\sigma \circ R_\pi$ for any $\pi, \sigma \in \mathbb{S}_m$. In the case that the $\{R_\pi\}$ maps are linear and represented by matrices, this reduces to $R_{\sigma\pi} = R_\sigma R_\pi$, which is equivalent to saying that they form a group representation of $\mathbb{S}_m$. ∎

**Proof of Proposition 4.** Under the action of a permutation $\pi \in \mathbb{S}_m$, $A$ and $B$ transform as

$$A \mapsto A' \qquad [A']_{j_1, \ldots, j_k} = [P_\pi]_{j_1}^{j'_1} [P_\pi]_{j_2}^{j'_2} \ldots [P_\pi]_{j_k}^{j'_k} [A]_{j'_1, \ldots, j'_k}, \tag{3}$$

$$B \mapsto B' \qquad [B']_{j_1, \ldots, j_p} = [P_\pi]_{j_1}^{j'_1} [P_\pi]_{j_2}^{j'_2} \ldots [P_\pi]_{j_p}^{j'_p} [B]_{j'_1, \ldots, j'_p}. \tag{4}$$

**Case 1.** Let $C = A \otimes B$. Under (3) and (4), $C$ transforms into

$$[C']_{i_1, \ldots, i_{k+p}} = \left([P_\pi]_{i_1}^{i'_1} \ldots [P_\pi]_{i_k}^{j'_k} [A]_{i'_1, \ldots, i'_k}\right) \left([P_\pi]_{i_{k+1}}^{i'_{k+1}} \ldots [P_\pi]_{i_{k+p}}^{i'_{k+p}} [B]_{i'_{k+1}, \ldots, i'_{k+p}}\right)$$

$$= [P_\pi]_{i_1}^{i'_1} \ldots [P_\pi]_{i_{k+p}}^{i'_{k+p}} C_{i'_1, \ldots, i'_{k+p}},$$

therefore, $C$ is a $k+p$'th order $P$–tensor.

**Case 2.** Let $C = A \odot_{(a_1, \ldots, a_p)} B$. Under (3) and (4), $C$ transforms as

$$[C']_{i_1, \ldots, i_k} = \left([P_\pi]_{i_1}^{i'_1} \ldots [P_\pi]_{i_k}^{i'_k} [A]_{i'_1, \ldots, i'_k}\right) \left([P_\pi]_{i_{a_1}}^{i'_{a_1}} \ldots [P_\pi]_{i_{a_p}}^{i'_{a_p}} [B]_{i'_{a_1}, \ldots, i'_{a_p}}\right) =$$

$$= [P_\pi]_{i_1}^{i'_1} \ldots [P_\pi]_{i_k}^{i'_k} \cdot [P_\pi]_{i_{a_1}}^{i'_{a_1}} \ldots [P_\pi]_{i_{a_p}}^{i'_{a_p}} \cdot [C]_{i'_1, \ldots, i'_k}.$$

Note that each of the $[P_\pi]_{i_{a_j}}^{i'_{a_j}}$ factors in this expression repeats one of the earlier appearing $[P_\pi]_{i_1}^{i'_1}, \ldots, [P_\pi]_{i_k}^{i'_k}$ factors, but since $P_\pi$ only has zero and one entries $[P_\pi]_{a,b}^2 = [P_\pi]_{a,b}$, so these factors can be dropped. Thus, $C$ is a $k$'th order $P$–tensor.

**Case 3.** Let $C = A\!\downarrow_{a_1, \ldots, a_p}$ and $b_1, \ldots, b_{k-p}$ be the indices (in increasing order) that are **not** amongst $\{a_1, \ldots, a_p\}$. Under (3), $C$ becomes

$$[C']_{i_{b_1}, \ldots, i_{b_{k-p}}} = \sum_{i_{a_1}} \cdots \sum_{i_{a_p}} [P_\pi]_{i_1}^{i'_1} \ldots [P_\pi]_{i_k}^{i'_k} [A]_{i'_1, \ldots, i'_k}$$

$$= [P_\pi]_{i_{b_1}}^{i'_{b_1}} \ldots [P_\pi]_{i_{b_{k-p}}}^{i'_{b_{k-p}}} \sum_{i'_{a_1}} \cdots \sum_{i'_{a_p}} [A]_{i'_1, \ldots, i'_k}$$

Thus, $C$ is a $k - p$'th order $P$–tensor.

**Case 4.** Follows directly from 3.

**Case 5.** Finally, if $A_1, ..., A_u$ are $k$'th order $P$–tensors and $C = \sum_j \alpha_j A_j$ then

$$[C']_{i_1,...,i_k} = \sum_j \alpha_j [P_\pi]_{i_1}^{i'_1} \ldots [P_\pi]_{i_k}^{i'_k} [A'_j]_{i'_1,...,i'_k} = [P_\pi]_{i_1}^{i'_1} \ldots [P_\pi]_{i_k}^{i'_k} \sum_j \alpha_k [A'_j]_{i'_1,...,i'_k},$$

so $C$ is a $k$'th order $P$–tensor. ∎

**Proof of Proposition 5.** Under the action of a permutation $\pi \in \mathbb{S}_{m'}$ on $\mathcal{P}_b$, $\chi$ (dropping the $^{a \to b}$ superscipt) transforms to $\chi'$, where $\chi'_{i,j} = \chi_{\pi^{-1}(i),j}$. However, this can also be written as

$$\chi'_{i,j} = [P_\pi \chi]_{i,j} = \sum_{i'} [P_\pi]_{i,i'} \chi_{i',j}.$$

Therefore, $\widetilde{F}_{i_1,...,i_k}$ transforms to

$$\widetilde{F}'_{i_1,...,i_k} = \chi'^{j_1}_{i_1} \chi'^{j_2}_{i_2} \ldots \chi'^{j_k}_{i_k} F_{j_1,...,j_k} = [P_\pi]_{i_1}^{i'_1} \ldots [P_\pi]_{i_k}^{i'_k} \chi^{j_1}_{i'_1} \chi^{j_2}_{i'_2} \ldots \chi^{j_k}_{i'_k} F_{j_1,...,j_k},$$

so $\widetilde{F}$ is a $P$–tensor. ∎

**Proof of Proposition 6.** By Proposition 5, under the action of any permutation $\pi$, each of the $\widetilde{F}_{p_j}$ slices of $\overline{F}$ transforms as

$$[\widetilde{F}'_{p_j}]_{i_1,...,i_k} = [P_\pi]_{i_1}^{i'_1} \ldots [P_\pi]_{i_k}^{i'_k} [\widetilde{F}'_{p_j}]_{i_1,...,i_k}.$$

At the same time, $\pi$ also permutes the slices amongst each other according to

$$\overline{F}'_{i_1,...,i_k,j} = [\widetilde{F}_{p_{\pi^{-1}(j)}}]_{i_1,...,i_k} = \overline{F}'_{i_1,...,i_k,\pi^{-1}(j)}.$$

Therefore

$$\overline{F}'_{i_1,...,i_k,j} = [P_\pi]_{i_1}^{i'_1} \ldots [P_\pi]_{i_k}^{i'_k} [P_\pi]_j^{j'} \overline{F}_{i_1,...,i_k,j},$$

so $\overline{F}$ is a $k+1$'th order $P$–tensor. ∎

**Proof of Proposition 7.** Under any permutation $\pi \in \mathbb{S}_m$ of $\mathcal{P}_i$, $A{\downarrow}_{\mathcal{P}'_i}$ transforms to $A{\downarrow}_{\mathcal{P}'_i}$, where $[A{\downarrow}_{\mathcal{P}'_i}]_{\pi(a),\pi(b)} = [A{\downarrow}_{\mathcal{P}_i}]_{a,b}$. Therefore, $A{\downarrow}_{\mathcal{P}_i}$ is a second order $P$–tensor. By the first case of Proposition 4, $F \otimes A{\downarrow}_{\mathcal{P}_i}$ is then a $k+2$'th order $P$–tensor. ∎

