# OpenReview forum: "Covariant Compositional Networks For Learning Graphs"
_ICLR.cc/2018/Conference — Invite to Workshop Track_

### Official Review · AnonReviewer2 · 2017-11-27
**Verbose paper that lacks lots of details on experiments**

**Rating:** 5
**Confidence:** 3

**Review:**

Thank you for your contribution to ICLR. The paper covers a very interesting topic and presents some though-provoking ideas.

The paper introduces "covariant compositional networks" with the purpose of learning graph representations. An example application also covered in the experimental section is graph classification.
Given a finite set S, a compositional network is simply a partially ordered set P where each element of P is a subset of S and where P contains all sets of cardinality 1 and the set S itself. Unfortunately, the presentation of the approach is extremely verbose and introduces old concepts (e.g., partially ordered set) under new names.  The basic idea (which is not new) of this work is that we need to impose some sort of hierarchical order on the nodes of the graph so as to learn hierarchical feature representations. Moreover, the hierarchical order of the nodes should be invariant to valid permutations of the nodes, that is, two isomorphic graphs should have the same hierarchical order on their nodes and the same feature representations. Since this is the case for graph embedding methods that collect feature representations from their neighbors in the graph (and where the feature aggregation functions are symmetric) it makes sense that "compositional networks" generalize graph convolutional networks (and the more general message passing neural networks framework).

The most challenging problem, however, namely the problem of finding a concrete and suitable permutation invariant hierarchical decomposition of the nodes plus some aggregation/pooling functions to compute the feature representations is not addressed in sufficient detail. The paper spends a lot of time on some theoretical definitions and (trivial) proofs but then fails to make the connection to an approach that works in practice. The description of the experiments and which compositional network is chosen and how it is chosen seems to be missing. The only part hinting at the model that was actually used in the experiments is the second paragraph of the section 'Experimental Setup', consisting of one long sentence that is incomprehensible to me.

Instead of spending a lot of effort on the definitions and (somewhat trivial) propositions in the first half of the paper, the authors should spend much more time on detailing the experiments and the actual model that they used. In an effort to make the framework as general as possible, you ended up making the paper highly verbose and difficult to follow.

Please address the following points or clarify in your rebuttal if I misunderstood something:

- what precisely is the novel contribution of your work (it cannot be "compositional networks" and the propositions concerning those because these are just old concepts under new names)?
- explain precisely (and/or more directly/less convoluted) how your model used in the experiments looks like; why do you think it is better than the other methods?
- given that compositional network is a very general concept (partially ordered set imposed on subsets of the graph vertices), what is the principled set of steps one has to follow to arrive at such a compositional network tailored to a particular graph collection? isn't (or shouldn't) that be the contribution of this work? Am I missing something?

In general, you should write the paper much more to the point and leave out unnecessary math (or move to an appendix).  The paper is currently highly inaccessible.

---

> ### Author Response · Authors · 2018-01-03
> **Reviewer has fundamentally misunderstood the paper, it is not surprising that the evaluation is negative**
>
> The reviewer's comments would be valid if our method really was based on imposing a partial order on the vertices of the input graph (i.e., if we were turning it into a DAG). However this is not what we do.
>
> Our algorithm operates with two separate graphs: the original graph G, and the composition scheme M constructed from G. M is indeed a DAG (i.e., it defines a partial order), but G is whatever the input is, so there is no need to impose "some sort of hierarchical order" on its nodes that the reviewer is missing from the presentation.
>
> The nodes of M correspond to *subsets* of the nodes of G, specifically neighborhoods of increasing radii, which naturally form a partial order by inclusion. This is explicitly stated in the paper in multiple places, eg., "In composition networks for graphs, the atoms will usually be the vertices, and the P_i parts will correspond to clusters of nodes or neighborhoods of different radii" (p.4). The construction of M is given explicitly on in points M1-M3 on page 5. Furthermore, Figure 5 shows how the neighborhoods are nested in the case of a simple input graph.
>
> The reason for all the definitions in Sections 3 and 4 is that we need to define how M is constructed and what covariance conditions the corresponding activations must satisfy. Incidentally, this is also the main novelty of the paper. If the reviewer didn't understand these sections, it is not surprising that the "Experiments" section seems seems hazy and the tensor contractions in Section 5 just seem like unnecessary fluff.
>
> Given the above misunderstanding it is also not surprising that the reviewer thinks that the paper is not very novel. The word "compositional network" may have been used before in different contexts, but the general way to construct a DAG from a structured input object, and the invariance properties discussed in Section 4 have not been discussed before in the literature. As we explain, some existing networks are special cases, but the general framework is entirely novel.
>
> By breaking down our presentation into a sequence of fairly precise definitions and propositions, and drawing figures such as Figure 2, which depicts the composition scheme, and Figure 5, which depicts the neighborhoods in G that correspond to each of the nodes of the composition scheme, we tried to make the notion of composition scheme as clear as possible. We are disappointed that the message still didn't get through, and we would welcome suggestions on how to better convey what is really the main message and main novelty in the paper. (Maybe by a figure with the original graph and the composition scheme side by side?)
>
> We would appreciate it if the reviewer revised his evaluation in light of the above clarification and would very much welcome suggestions on how to change the presentation so as to avoid other leaders falling prey to the same misunderstanding. However, we can vouch that the math is not unnecessary: without it, this construction would simply not work.

---

> > ### Comment · AnonReviewer2 · 2018-01-03
> > **Clarification**
> >
> > Before responding to your rebuttal, could you please point out precisely where I "fundamentally misunderstood" your paper. You do impose a partial order on the nodes of the input graph. That's not what I could have misunderstood. Thanks!

---

> > > ### Author Response · Authors · 2018-01-03
> > > **Clarification**
> > >
> > > You fundamentally misunderstood our paper because you think we impose a partial order on the nodes of the input graph. We do not. I cannot stress this enough.
> > >
> > > There are two different graphs in the paper: the input graph G and the corresponding composition scheme M. G is an an undirected graph and there is no partial order on it. M is a directed acyclic graph (by construction) and therefore it defines a partial order.
> > >
> > > I can see where you are coming from because if there was a partial order on the nodes of G in the first place then it would be easier to turn it into a neural network. That would be a "cheap" way of creating a representation for G. However, as you rightfully point out, imposing a partial order on the nodes of undirected graph so as to reflect multiscale structure is a thorny problem that we explicitly want to avoid (there is a literature on graph reductions which gets tangled up in exactly this issue).
> > >
> > > A lot of the paper (specifically, Section 3) is concerned with the relationship between G and M. Some of what you deem "unnecessary math" is about how to construct M in such a way that it behaves appropriately under transformations of G (Figure 3).
> > >
> > > We do appreciate your quick response because it is essential that we clarify this point. I would also like to understand what it is that made you think that we impose a partial order on the nodes of G, because if you were mislead by this, then other readers will be mislead too. Turning your question around, can you point out the specific location in the paper that made you think that we impose a partial order on G?
> > >
> > > Once again, G and M are different graphs. Every node of M corresponds to a set of nodes of G. Typically, M has many more nodes than G. There is no partial order on the nodes of G.

---

> > > > ### Comment · AnonReviewer2 · 2018-01-16
> > > > **Clarification**
> > > >
> > > > Dear authors,
> > > >
> > > > Thanks for the additional information.
> > > >
> > > > When I write that a partial order is imposed on the nodes of the graph, I mean a partial order where the elements that are ordered are subsets of the set of nodes of the input graph G. This is indeed the case, because in the graph M the nodes are associated with subsets of nodes of the graph G (each node in M is associated with such a subset = the receptive field). I do understand  now, however, that my words could be understood as me saying that your approach imposes an order on individual nodes of G. I know this is not the case.
> > > >
> > > > Let me take a step back and let me try to explain again why I think that some parts of the paper are unnecessarily verbose and make it much less accessible than it could be.
> > > >
> > > > First, let me reformulate the computational problem you are addressing here (as far as I understand it).
> > > >
> > > > The input is a graph G (let's call this the data graph) and the output is a computation graph M. The computation graph M has several properties:
> > > >
> > > > (1) The nodes are associated with computations that are determined by the aggregation functions. Incoming edges determine the input to the performed computation (the variables).
> > > > (2) The nodes are associated with a receptive field P. This shows which basic variables (features at individual nodes of G) are involved in the operation that's performed at the node of M.
> > > > (3) The nodes in the first layer are associated with the nodes of G. This makes sense because we want to always start with the nodes in G and their features. The function with these features as variables is determined by the computation graph M.
> > > > (4) We want M to be invariant to (certain types of) symmetries of G. For instance, some permutation of G's labeling that is an isomorphism should also be an isomorphism in M. (Wrt to the graph structure but also the aggregation functions that are computed.)
> > > >
> > > > Again, this is the computational problem as I understand it. And it is a problem that everyone who tries to learn NNs for graphs is (implicitly) working on.  Everything you write in section 3 is a (in my opinion) verbose way of setting up this problem. Algorithm 1 is one way of creating M from G commonly used in the literature. It simply uses the neighborhood information to determine the edge structure of M.
> > > >
> > > > My worry about your paper is that the reader is already lost at a point where you have essentially just reformulated a problem that several papers have addressed before.
> > > >
> > > > The interesting new parts of the paper are in section 4 and 5. Unfortunately, I think that these sections are also in parts unnecessarily  technical and lack intuitive examples. I generally appreciate works that provide a unifying framework capturing previous work as specific instances. I think that the paper succeeds here and that this is the major contribution. But I also think that it is the authors job to make the paper accessible. That's the reason why I would urge the authors to rework the presentation. Add more examples (the ones given are not very helpful), describe the intuitions behind the definitions earlier etc.
> > > >
> > > > Finally, while the paper is exhaustive on the technical parts, it provides very little details on the experimental set up. There is one short paragraph explaining your model. It would be very helpful to the reader if you would explain the links between the model used in your experiments and the theory introduced earlier in more detail.
> > > >
> > > > in summary, I appreciate the effort of the authors to engage in a conversation that did help me to better understand the paper. In light of this discussion, I'll increase my score by one point. I am still tending more towards a rejection because of the papers presentation that could be improved substantially and also due to the lack of details in the experimental section.

---

> > > ### Author Response · Authors · 2018-01-03
> > > **Algorithm clarification - Part 1**
> > >
> > > Dear Reviewer 2,
> > >
> > > Thank you very much for your effort in understanding our paper. I think it is necessary for us to clarify our algorithm here again without too much mathematics to avoid further misunderstanding. Please check my following words carefully.
> > >
> > > (1) Based on our definition, all other graph neural networks in the current literature, in particular Neural Graph Fingerprint [Duvenaud et al, 2015], Graph Convolution Neural Network [Kipf et al, 2016], Gated Graph Sequence Neural Network [Li et al, 2016], Learning Convolution Neural Network for Graphs [Niepert et al, 2016], are classified as the "zero-order message passing" in which at any level/layer/iteration the vertex representation is a vector, each element of the vector is for a channel. In another words, every channel is represented by a scalar.
> > >
> > > (2) From (1), we realize that to empower the representation we need more than a scalar per channel. We introduce the "first-order message passing" or in another name the "first-order covariant compositional network" in which the vertex representation of vertex v is a matrix, each row of the matrix corresponds to a vertex w in the receptive field of the vertex v. Now we can see that each channel (column of the matrix) is represented by a vector with the length as the size of the receptive field of v. At level/layer/iteration 0, the receptive field of v is a set containing only v. The receptive field of v grows gradually in the following level/layer/iteration.
> > >
> > > (3) Furthermore, we introduce the "second-order message passing" or in another name the "second-order covariant compositional network" such that the vertex representation of vertex v is now a 3-order tensor in which each channel is represented by a matrix of size N x N where N is the size of the receptive field of v.
> > >
> > > (4) You may ask a question: The receptive field of v or the set of vertices in the extended neighborhood of v can appear in any order, we have to use some algorithm to find the "partial ordering" of that receptive field? The answer is we do NOT need such algorithm. If we use some algorithm like Weisfeiler-Lehman algorithm to rank the vertices then it becomes Learning Convolution Neural Network for Graphs [Niepert et al, 2016]. In addition, finding an optimal ordering is an NP-hard problem, the Weisfeiler-Lehman isomorphism test still fails with a very small probability. You can check [Babai 2015] https://arxiv.org/pdf/1512.03547.pdf for details. What we want is a neural network model that is perfectly permutation-invariant.
> > >
> > > [TO BE CONTINUED]

---

> > > ### Author Response · Authors · 2018-01-03
> > > **Algorithm clarification - Part 2**
> > >
> > > [CONTINUE FROM PART 1]
> > >
> > > (5) To address the question from (4) in designing a perfectly permutation-invariant neural network, here is the algorithm.
> > >
> > > First, consider a level/layer/iteration L-th of the message passing. We compute the receptive field of v at level L as the union of the receptive fields at level L - 1 of vertices w in the neighborhood of v. Take the "first-order covariant compositional network" as an example where the vertex representation is a matrix. The number of rows of vertex representation of w at level L - 1 is smaller or equal to the number of rows of vertex representation of v at level L, because the receptive field of w at level L - 1 is a subset of the receptive field of v at level L. We need to have the sizes of these two matrices equal. We can do so by multiplying the vertex representation of w at level L - 1 with a permutation matrix P on the left side. This permutation matrix P is simply defined as follows: P_ij = 1 if the i-th vertex in the receptive field of v at level L is the j-th vertex in the receptive field of w at level L - 1. In the case of "second-order covariant compositional network", we have to broadcast-multiply P and P-transpose on the left and right hand side.
> > >
> > > Second, now all the vertex representations of vertices in the receptive field of v have the same size. We concatenate or formally saying "tensor stack" all these vertex representations. We obtain a higher-order tensor. From here, as in the "second-order covariant compositional network", we can "tensor product" this higher-order tensor with the reduced adjacency matrix of v at level L (subject to the receptive field of v at level L) and obtain an even higher-order tensor. For example, in the second-order CCN, after the "tensor product" step, we obtain a 6-order tensor.
> > >
> > > Third, given a high-order tensor as the representation of vertex v, we have to "tensor contract" or "tensor reduce" or in normal word "shrink" back it into a lower-order tensor for feasible computation. These tensor contraction operations as we define are perfectly permutation-invariant. Thus, we contract from high-order tensor into a matrix (in first-order CCN) or in a 3-order tensor (in second-order CCN).
> > >
> > > Forth, we apply a learnable weight for all the channels in all vertex representations after the tensor contraction step. These learnable weights are leanred via back-propagation.
> > >
> > > Fifth, on top of the network, we again "shrink" the vertex representations (matrices in first-order CCN, or 3-order tensors in second-order CCN) into vectors of channels. We sum up all these "shrinked representations" into a single vector, this is the vector for further regression/classification task. In addition, we can concatenate all "shrinked representations" of all levels, this can be a richer graph representation.
> > >
> > > (6) You may ask a question: The high-order tensors are huge, how can we deal with them? The answer is: we do NOT do the "tensor product" operation explicitly, because it cannot be hold in the computer memory. In the "tensor contraction" step, for example with GPUs, we introduce a "virtual indexing system" for a "virtual tensor" that computes the element of tensor only when needed given the index.
> > >
> > > Thank you so much for your consideration. Please let us know your further questions. We look forward to hear from you soon.
> > >
> > > Best regards,
> > > Representative of the paper authors

---

### Official Review · AnonReviewer1 · 2017-11-27
**The paper presents a generalized architecture for representing generic compositional objects, such as graphs, which is called covariant compositional networks (CCN).**

**Rating:** 5
**Confidence:** 2

**Review:**

The paper presents a generalized architecture for representing generic compositional objects, such as graphs, which is called covariant compositional networks (CCN). Although, the paper is well structured and quite well written, its dense information    and its long size made it hard to follow in depth. Some parts of the paper should have been moved to appendix. As far as the evaluation, the proposed method seems to outperform in a number of tasks/datasets compare to SoA methods, but it is not really clear whether the out-performance is of statistical significance. Moreover,  in Table 1, training performances shouldn't be shown, while in Table 3, RMSE it would be nice to be shown in order to gain a complete image of the actual performance.

---

> ### Author Response · Authors · 2018-01-03
> **Please reflect on the entire paper, not just the "Experiments" section**
>
> Thanks for your comments. The paper is longer than usual because rather than just proposing a tweak on some existing algorithm, it derives a general framework for dealing with the issue of covariance in compositional-type neural architectures. We couldn't figure out how to present all this in less than 15 pages without making the paper unreadably dense.
>
> Our results on HCEP are quite spectacular. While we didn't perform formal tests of statistical significance, it is quite clear that CCN far outperforms the other methods. For the MUTAG, etc. benchmarks it is harder to show statistical significance, simply because these datasets are small and multiple algorithms are neck and neck. For QM9, although PSCN is admittedly close, we consistently beat it (as well as the other two competitors) on all 13 regression tasks in both MAE and RMSE, which again gives us confidence that the results are not just a statistical fluke.
>
> Thank you for your suggestion to include RMSE, we added a separate table to show that. Incidentally, the experiments required writing our own custom deep learning library in C++ (which now can also use GPUs). The code has been released on GitHub, unfortunately we cannot provide a link at this point because it would break the anonymity of the submission.
>
> Notwithstanding the above, we feel that sate-of-the-art experimental results are just one part of the paper's contribution. The new conceptual framework of compositional networks and our mathematical results on how to make them covariant is even more important. We would appreciate it if the review was revised to reflect on the main body of the paper as well, not just the "Experiments" section.

---

### Official Review · AnonReviewer3 · 2017-11-27
**Nice technical contribution with some presentation issues**

**Rating:** 6
**Confidence:** 3

**Review:**

The paper introduces a formalism to perform graph classification and regression, so-called "covariant compositional networks", which can be seen as a generalization of the recently proposed neural message passing algorithms.

The authors argue that neural message passing algorithms are not able to sufficiently capture the structure of the graph since their neighborhood aggregation function is permutation invariant. They argue that relying on permutation  invariance will led to some loss of structural information.

In order to address this issue they introduce covariant comp-nets, which are a hierarchical decompositon of the set of vertices, and propose corresponding aggregation rules based on tensor arithmetic.

Their new method is evaluated on several graph regression and classification benchmark data sets, showing that it improves the state-of-the-art on a subset of them.

Strong points:
+ New method that generalizes existing methods

Weak Points:
- Paper should be made more accessible, especially pages 10-11
- Should include more data sets for graph classification experiments, e.g., larger data sets such as REDDIT-*
- Paper does not include proofs, should be included in the appendix
- Review of literature could be extended

Some Remarks:
* Section 1: The reference Feragen et al., 2013 is not adequate for kernels based on walks.
* Section 3 is rather lengthy. I wonder if its contents are really needed in the following.
* Section 6.5, 2nd paragraph: The sentence is difficult to understand. Moreover, the reference (Kriege et al., 2016) appears not to be adequate: The vectors obtained from one-hot encodings are summed and concatenated, which is different from the approach cited. This step should be clarified.
* unify first names in references (Marion Neumann vs. R. Kondor)
* P. 5 (bottom) broken reference

---

> ### Author Response · Authors · 2018-01-03
> **Fair review**
>
> Thank you for your review. We tried to fix everything that you mention. In particular: we have added an appendix that contains all the proofs; added one more reference for kernels based on counting walks; rewrote the paragraph you mention in the "Experiments" section; generally extended and cleaned up the experimental results.
>
> We appreciate your comment about Section 3. The reason that we structured the paper as we did was to emphasize that we are have a new general architecture for learning from structured (multiscale) objects and that graphs are just a special case. In fact, this work started out significantly more abstract and general, revolving around ideas from representation theory, and we were pretty happy that by reformulating it in the compositional framework we could condense it to something that can be described in 15 pages and ultimately reduces to just tensor products and contractions. It seems like maybe we are not doing a very good job of conveying the generality and power of the approach, though, because all three reviews complain about "why don't you just get down to describe your graph algorithm?". Pages 10-11 may be dense, but the truth is that, at the end of the day, the tensor operations that they describe are mercifully straightforward, almost trivial.
>
> Having said this, we could not find any existing deep learning software that implements the type of contractions that we need, so we had to write our own library. The library has now been released, but we cannot include a link here because it would break the anonymity of the submission. At the time of the submission the library could only use CPUs, that's why the datasets are relatively small. Since then, we have extended the library to be able to use GPU's so we now have the capability of running larger experiments and we will definitely try REDDIT. Thanks for the suggestion.

---

### Public Comment · (anonymous) · 2017-12-05
**Related Work**

The graph kernel benchmark has two more datasets (Proteins and Enzymes), it would be interesting if the authors can report the results on them. Also, there are some other papers like Deep Graph Kernels (KDD 2015) and Optimal Assignment WL-graph kernel (NIPS 2016), that their results are not mentioned in table 2.

---

> ### Public Comment · (anonymous) · 2017-12-11
> **Other data sets**
>
> Social media data-sets, e.g. REDDIT-* and IMDB-BINARY, should also be interesting. Have a look at data sets used in the OA-Kernel-Paper or graphkernels.cs.tu-dortmund.de.

---

### Public Comment · ~Justin_Gilmer1 · 2017-12-17
**Question regarding invariance**

Hello, nice paper! I have a question about the following claim - "While MPNNs have been very successful in various applications and are an active field of research, they differ from classical CNNs in a fundamental way: the internal feature representations in CNNs are equivariant to transformations of the input such as translation and rotations (Cohen & Welling, 2016a;b), contrasted with those in MPNNs, which are merely invariant."

The MPNN message passing phase is equivariant to permutations of the nodes, it is only the readout phase which is invariant. Am I missing something here?

---

> ### Author Response · Authors · 2017-12-17
> **Invariance**
>
> Dear Justin,
>
> Thanks for reading and your comment!
>
> In section 4, we explicate on this a bit more. The output of an internal node in a MPNN depends on its receptive field as a set (and not an ordered set), so it is invariant to permutations of the nodes, not covariant.
>
> I hope that answers your question?

---

### Decision · Program_Chairs · 2018-01-29
**ICLR 2018 Conference Acceptance Decision**

**Decision:**

Invite to Workshop Track

**Comment:**

This is a good contribution, with the potential to become extremely good and significant if presentation is substantially improved.
All reviewers comment on the lack of clarity of the paper, especially concerning its central contributions (Section 4 and 5), as illustrated also by the relatively low confidence scores.
Reviewers also mention the current imbalance between the generality of high-order compositional networks and the motivation and empirical evaluation of these models. Generalizations of graph neural representations based on higher order local interactions are particularly interesting in contexts such as combinatorial optimization, where heuristics typically exploit high-order interactions.

In summary, we believe this work deserves a further iteration before it can be in proceedings in order to improve the exposition and the motivation of compositional networks, that will greatly improve its exposure to the community.  That said, the idea it lays forward is of potential interest, and thus the AC recommends resubmission to the workshop track.